# Increase in antioxidant capacity associated with the successful subclone of hypervirulent carbapenem-resistant *Klebsiella pneumoniae* ST11-KL64

Ruobing Wang ®[1], Anru Zhang[1], Shijun Sun[1], Guankun Yin[1], Xingyu Wu[1], Qi Ding[1], Qi Wang[1], Fengning Chen[1], Shuyi Wang[1], Lucy van Dorp ®[2], Yawei Zhang[1], Longyang Jin[1], Xiaojuan Wang[1], Francois Balloux ®[2] & Hui Wang ®[1] ✉

The acquisition of exogenous mobile genetic material imposes an adaptive burden on bacteria, whereas the adaptational evolution of virulence plasmids upon entry into carbapenem-resistant *Klebsiella pneumoniae* (CRKP) and its impact remains unclear. To better understand the virulence in CRKP, we characterize virulence plasmids utilizing a large genomic data containing 1219 *K. pneumoniae* from our long-term surveillance and publicly accessible databases. Phylogenetic evaluation unveils associations between distinct virulence plasmids and serotypes. The sub-lineage ST11-KL64 CRKP acquires a pK2044-like virulence plasmid from ST23-KL1 hypervirulent *K. pneumoniae*, with a 2698 bp region deletion in all ST11-KL64. The deletion is observed to regulate methionine metabolism, enhance antioxidant capacity, and further improve survival of hypervirulent CRKP in macrophages. The pK2044-like virulence plasmid discards certain sequences to enhance survival of ST11-KL64, thereby conferring an evolutionary advantage. This work contributes to multifaceted understanding of virulence and provides insight into potential causes behind low fitness costs observed in bacteria.

The increasing global prevalence of carbapenem-resistant *Klebsiella pneumoniae* (CRKP) poses a major challenge to human health and antimicrobial stewardship efforts. In Asia[1], sequence type (ST) 11 is the dominant clone, responsible for infections correlated with a high in-hospital mortality rate of 33.5%[2]. Of concern is the emergence of hypervirulent carbapenem-resistant *K. pneumoniae* (hv-CRKP), which can cause hard to treat infections with poor outcomes. In particular, ST11-KL64 CRKP had a higher incidence of sepsis, which served as an independent risk factor for mortality[3]. Over the past decade, hv-CRKP has rapidly disseminated, a phenomenon typically attributed to the acquisition of virulence plasmids containing prevalent virulence genes, such as *iucA*, *iroN*, *rmpA*, and *rmpA2*[4].

Despite virulence plasmid acquisition, not all hv-CRKP exhibit phenotypic traits similar to hypervirulent *K. pneumoniae* (hvKP), such as mucoviscosity and siderophore production. The inconsistency between phenotypes and genotypes of ST11 hv-CRKP in virulence characterizations has been previously observed[5], pointing the effects of virulence plasmid acquisition meriting further investigation.

Plasmids are regarded as conferring new adaptive capabilities to bacteria while also imposing a fitness cost. Adaptive evolution, driven by mutations linked to oxidative stress, nucleotide synthesis, and metabolism, alleviates the costs associated with plasmid carriage, facilitating widespread dissemination[6,7]. Therefore, investigating the adaptive evolution of bacteria following the acquisition of virulent

[1]Department of Clinical Laboratory, Peking University People's Hospital, Beijing, People's Republic of China. [2]UCL Genetics Institute, Department of Genetics, Evolution & Environment, University College London, London, UK. ✉e-mail: whuibj@163.com

plasmids is essential for comprehending the potential prevalence of hv-CRKP.

Prior work has focused on the molecular epidemiology, virulence, and antimicrobial resistance of hv-CRKP, including sporadic cases of genomic evolution of hv-CRKP[4,8–12]. However, to date, the complexities of fitness costs and compensatory evolution that reduces the burden of plasmid carriage remain partially understood.

Here, we investigate the genomic evolution of virulence plasmids and observed a specific conserved deletion in the pK2044-like plasmid when in the ST11-KL64 sub-lineage. We predict the function of the specific deletion in genome and transcriptome levels, and provide experimental evidence demonstrating that adaptive alterations in virulence plasmids increase the bacterial survival rate by amplifying the antioxidant capacity of hv-CRKP. Our research indicates that the pK2044-like plasmid confers evolutionary benefits to ST11-KL64 by discarding certain sequences, which offers insight into the widespread prevalence of ST11-KL64 CRKP in severe infections.

## Results

### The genome dataset of the predominant clone ST11 *K. pneumoniae*

We curated a total of 1219 *K. pneumoniae* genome assemblies, including 302 newly generated assemblies, from isolates spanning 31 regions sampled between 2002 and 2019. A total of 744 isolates had assigned geographic information available (Supplementary Fig. 1a). Most isolates were from China (n = 403, including our newly generated data), followed by the United States (n = 175) and Switzerland (n = 60). Most

strains were isolated from humans (n = 646, 93.9%), while the rest were mainly sampled from livestock (n = 29) and clinical environments (n = 12).

The *K. pneumoniae* collection could be divided into 215 distinct STs (Supplementary Fig. 1b). The ST11 clone (20.8%, 254/1219) dominated, followed by ST258 (13.1%, 160/1219). Of the 254 ST11 assemblies, 228 isolates were from China (90.0%, Supplementary Fig. 1c). Beijing (n = 61), Zhejiang (n = 31), and Henan (n = 26) were the provinces or municipalities having the most ST11 isolates. 45.6% of ST258 isolates were collected from the United States (n = 73), however, none of these isolates harbored virulence plasmids based on the presence of virulence associated marker genes. Therefore, ST11 *K. pneumoniae* was chosen as the host strain for analysis of virulence plasmids in this study.

### Phylogeny of chromosome exhibit selective preference towards plasmids

A total of 246 assemblies were considered sufficiently complete scaffolds (longest contig > 4.5 Mb) to study the evolutionary history of the chromosome of ST11 *K. pneumoniae*. Recombinant regions were excluded from the analysis, and their positions can be found in Supplementary Fig. 2. A recombination filtered core genome phylogeny revealed several distinct phylogenetic clades (Fig. 1). Most of KL47 and KL64 isolates formed a monophyletic cluster, indicating a close genetic relationship. In contrast, other serotypes of ST11 isolates were distributed across different clades with a high genetic diversity, mostly originating from countries outside of China. Isolates within China also

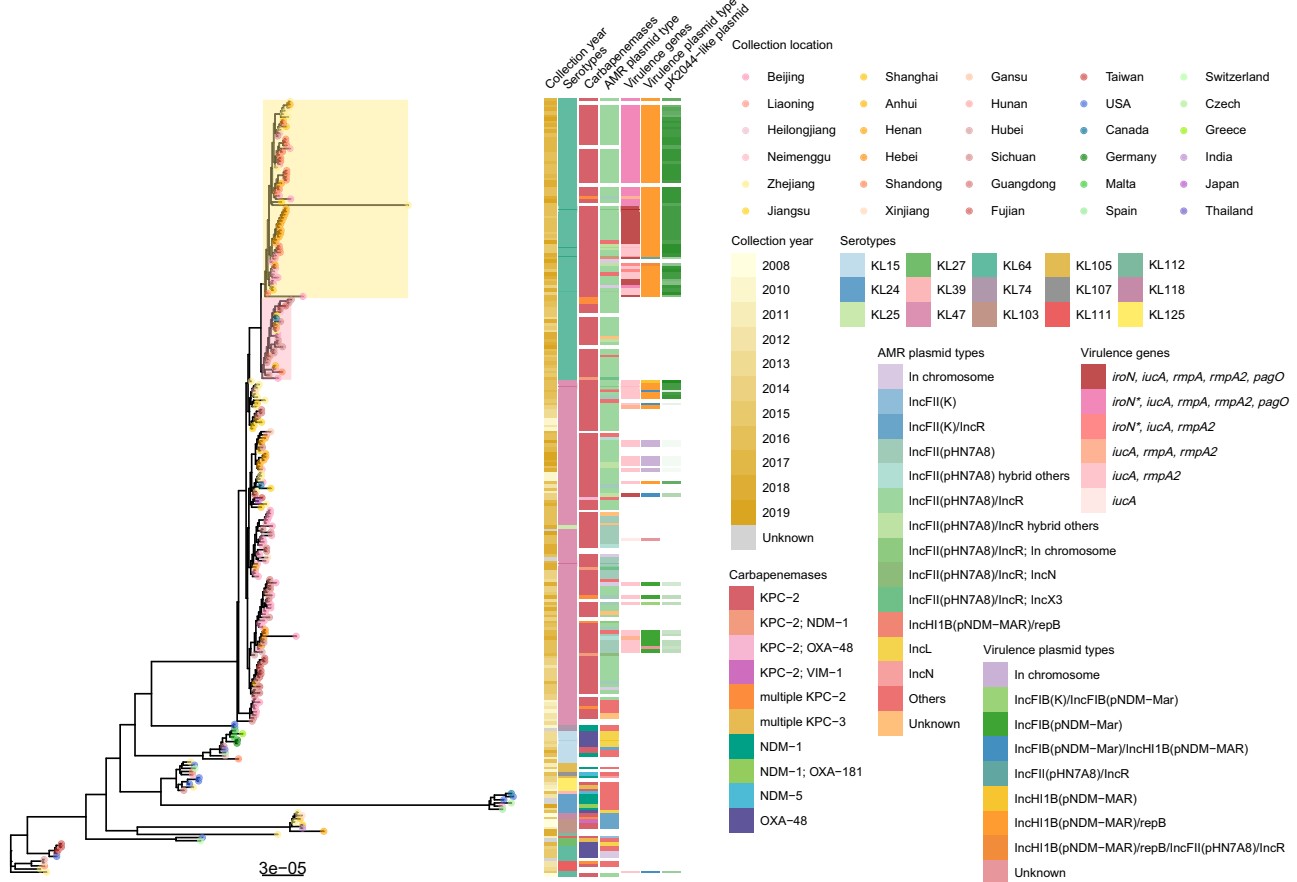

**Fig. 1 | Phylogenetic tree of ST11 CRKP chromosome.** Tip points of the phylogenetic tree are dotted by location where strains were isolated. Heatmap denotes (from left to right) collection date, serotypes, the presence of carbapenemase genes, types of carbapenem resistance plasmids, the presence of virulence genes, types of virulence plasmids and the similarity to pK2044. *iroN** is an incomplete gene, with only the first 1306 bp mapping out of a total length of 2181 bp. The similarity to pK2044 was measured by the coverage of comparison of pK2044 and each plasmid using BLASTn. Source data are provided as a Source Data file.

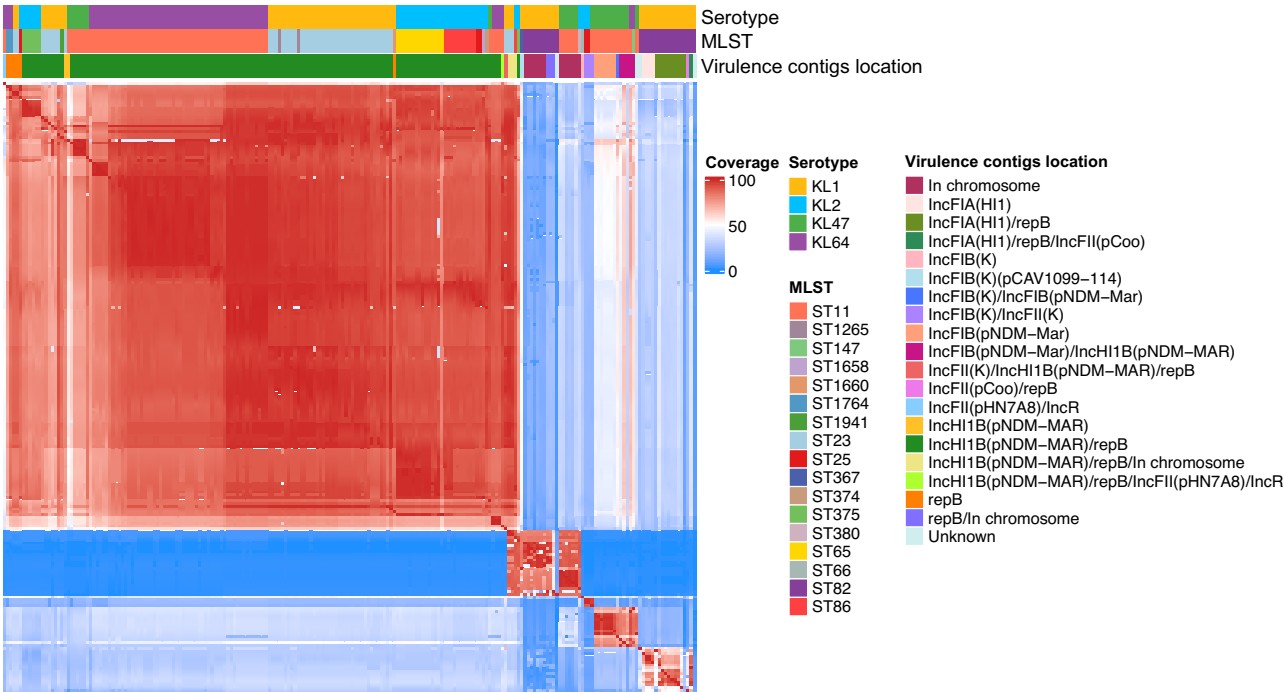

**Fig. 2 | Comparison of virulence contigs from four major serotype *K. pneumoniae.*** Each of 217 virulence contigs underwent pairwise by fastANI, and coverage (aligned/total) of comparison results is showed in the heatmap. The shade of each cell in the heatmap represents the coverage of the two virulence contigs on the corresponding *x* and *y* axes. The order of the virulence contigs on the *x* and *y* axes is consistent. Source data are provided as a Source Data file.

exhibited a pattern of regional clustering with the occurrence of genetically related isolates in the same or closely neighboring geographical location.

To reconstruct the phylogenetic history, we first excluded seven assemblies which were missing information on the date of collection. After testing temporal signal (Supplementary Fig. 3 & Supplementary Table 1), the multisequence alignment including 218 independent isolates spanning 1,341,180 sites was created for the final analysis. With a mutation rate of roughly $1.126 \times 10^{-6}$ substitutions per site per year [$0.743$–$1.543 \times 10^{-6}$ 95% highest poster density (HPD)], we inferred that the common ancestor for the ST11 *K. pneumoniae* chromosome was in 1891 (1864–1916, 95% HPD with a strict clock and coalescent constant model). Around 2000, a common ancestor of the prevalent ST11 strain in China emerged, followed by a divergence from a branch of the KL47 strain, resulting in the formation of KL64 (Supplementary Fig. 4).

Interestingly, CRKP and hv-CRKP belonging to the KL64 serotype were dispersed across two separate clades (colored in pink and yellow). Virulence plasmids were identified in nearly all strains within the yellow clades, sharing significant similarities to the classical virulence plasmids pK2044. In a specific clade within the yellow clade, virulence gene *iroN* is partial, with the first 1306 bp out of the total length of 2181 bp mapped. This finding suggests a potential preference of virulence plasmids for the ST11 host, particularly within a sublineage KL64.

### Virulence plasmids in ST11-KL64 derived from a sub-lineage of ST23-KL1

We identified virulence associated contigs based on the presence of the virulence associated genes *rmpA*, *rmpA2*, *iroN*, *iucA*, or *pagO* in KL1, KL2, KL47 and KL64 genomes because they are to the most prevalent serotypes in hypervirulent or ST11 *K. pneumoniae*. A total of 217 contigs were identified and classified into 17 plasmid replicon types. IncHI1B(pNDM-MAR)/repB (67.7%, 147/217) was the most common plasmid type, which was identical to the classic virulence plasmid pK2044. Despite in KL64, approximately 100 bp in the *ori* region

determining the IncHI1B plasmid type failed to align. Through pairwise comparisons, we observed that the set of virulence contigs is not highly similar (Fig. 2). Most of the virulence contigs from KL1, KL2 and KL64 strains showed high similarity (in red region), while certain contigs from KL1 and KL47 strains were substantially dissimilar.

We selected the 141 virulence contigs with high similarity (an Average Nucleotide Identity > 95 and coverage > 80%) to pK2044 for phylogenetic reconstruction (Fig. 3). It found that the virulence contigs in the same lineage were predominantly derived from strains of the same MLST or serotype. The pK2044-like plasmids in ST11-KL64 (clade in red) and a subset of ST23-KL1 exhibited a closer phylogenetic relationship, indicating that those plasmids may share a common ancestor. While the KL2 clade (clade in yellow) were highly divergent with the KL1 and KL64 clades. It could be boldly speculated that the virulence plasmid on ST11-KL64 was most likely directly evolved from sublineages of those on ST23-KL1.

### Specific genes of pK2044-like plasmid deletion in ST11 hv-CRKP

We attempted to identify specific genes within the pK2044-like plasmids that are unique to different hosts (with different ST or serotypes), aiming to investigate if there are host-specific variations on the virulence plasmids. Several specific genes were identified, though few coding for characterized proteins (Fig. 3 & Supplementary Table 2). For example, in ST11-KL64, the gene *g142* (957 bp) encoding the DsbA family protein, which was considered essential for bacterial virulence factor assembly. Gene *g536* (483 bp) and *ymoA*, encoding the GNAT family N-acetyltransferase and a global modulator controlling virulence gene expression[13], were also shown to be enriched in KL2 strains.

Specific genes, *g417* (513 bp), *g280* (516 bp), and *g499* (603 bp), were found to be strongly associated between the virulence plasmid of KL1 strains belonged to a 2698 bp region spanning 69,203-71,901 of pK2044 (marked with a red box in Fig. 4a). This region was identified as fully absent in the virulence plasmid in association with all ST11-KL64. Such an observation is consistent with either i) a founder event, where the pK2044-like plasmid originally acquired by ST11-KL64 exhibited

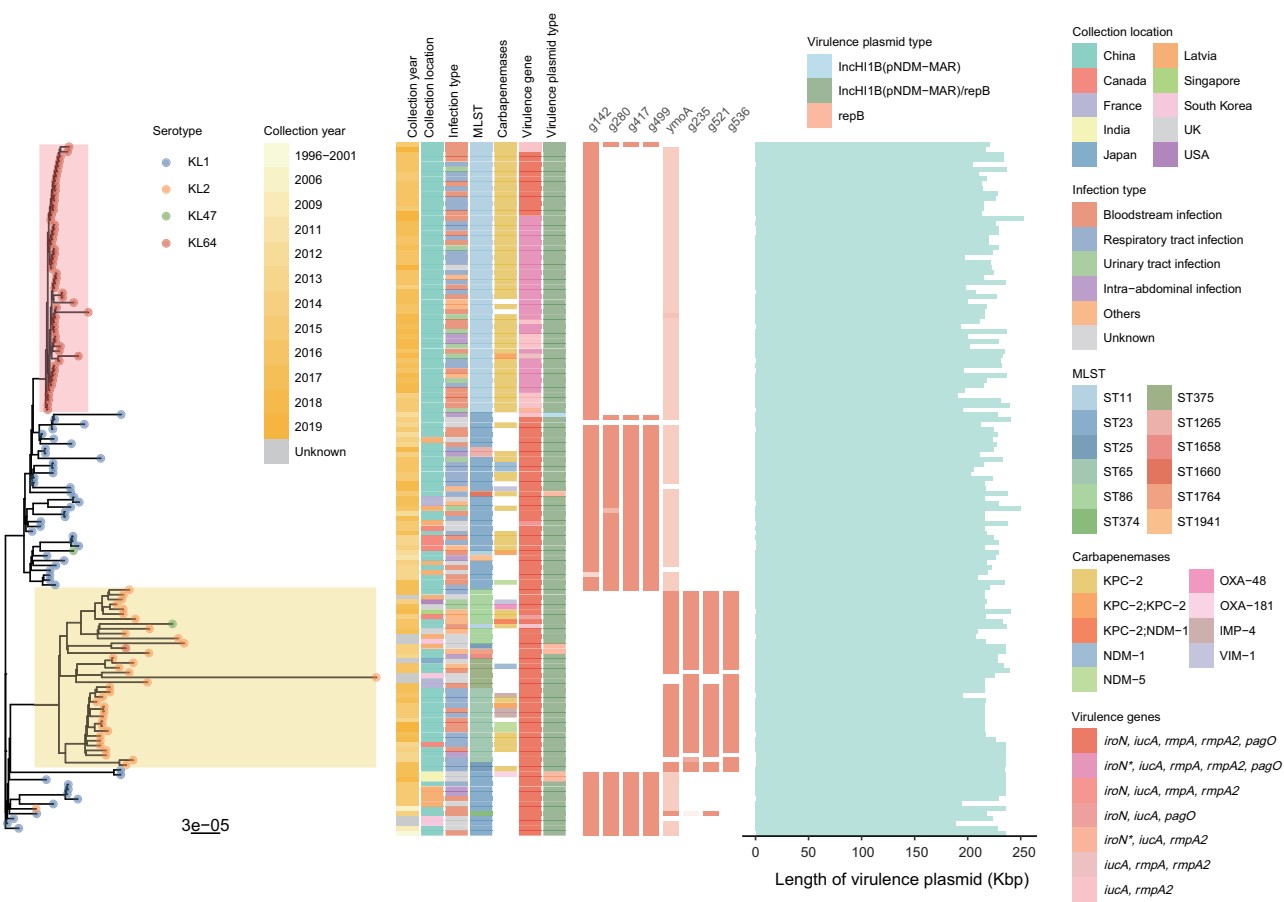

**Fig. 3 | Phylogenetic tree of virulence plasmids.** Main clades containing several stains are colored in red (clade KL64) and yellow (clade KL2). Tips the phylogenetic tree are colored by serotype of the host strains. Heatmap denotes (from left to right) collection date, regions of collection, types of the host infection, MLST, the presence of carbapenemase genes, the presence of virulence genes, types of virulence plasmids, and the presence of specific genes on virulence plasmids for variant serotypes of strains. Bar chart shows length of virulence plasmids. Visualized with pheatmap v1.0.12 package in R v4.3.1 (https://www.r-project.org/). Source data are provided as a Source Data file.

this deletion and subsequently persisted in circulating strains or ii) an adaptation to colonization and maintenance in an ST11-KL64 host.

We note however that some deletion is a feature of the ST11-KL64 pK2044-like plasmid we characterize. For example, a portion of *repB* (97 of 570 bp), which defines the IncHI1B(pNDM-MAR) replicon type spanning roughly 150 Kb of pK2044, was also absent in the genome of all ST11-KL64 virulence plasmids. Furthermore, over two-thirds (69.4%, 43/62) of ST11-KL64 virulence plasmids missed a ~16 Kb region located at positions 13,666–29,915. This region contains the salmochelin related gene *iroBCDN*, ferric citrate transport activate gene *fecI, fecR, fecA*, phosphoadenosine phosphosulfate reductase *cysH* and transposase genes. The region was also either completely or partly absent in most of the ST11-KL47 strains studied (96.3%, 26/27). The absence of this region is a possible contributor to the reduced virulence phenotype observed in ST11 hv-CRKP compared to ST23 hvKP.

**The deletion region was predicted to generate enzymes with activity**

Three complete proteins Hp2 (encoded by *g280*), Hp3 (encoded by *g499*), Hp4 and two truncated hypothetical proteins Hp1 (encoded by *g417*) and Hp5 were predicted within the ~3 Kb region, although Hp4 was too short and not consistently annotated (Fig. 4b). For comparison, AT content was 64.3%, 66.7%, and 61.0% in *hp1* to *hp3*, respectively, implying the region may be regulated by a global transcriptional repressor H-NS and its family protein[14]. To investigate the function of these genes we started by predicting protein structure models (see

Methods). Following application of Phyre2, Hp3 was predicted to be a phosphatase with 92.4% confidence and 52% coverage. Hp3 was found to be comparable to flavin mononucleotide (FMN) reductase (NADPH) in AlphaFold2 database based on its anticipated structure model [predicted Local Distance Difference Test (pLDDT) > 90]. The truncated protein Hp5 could be predicted as a stress response to the $H_2O_2$-related protein YaaA of *Escherichia coli* with 98.2% confidence, 64% coverage, and pLDDT > 92. Another truncated protein, Hp1, was indicated to be a membrane protein due to the four predicted helix bundles in its core area. More reliable structure models were not predicted for Hp2 and Hp4 (Supplementary Fig. 5).

**The deletion region affected the methionine metabolism on ST11 hv-CRKP**

Due to their representative pK2044-like virulence plasmids and KPC-2 plasmids, we selected strains of a ST11-KL64 hv-CRKP C1789 and a ST23-KL1 carbapenem resistant hvKP (CR-hvKP) C4599 to study other potential functions of the ~3 Kb deleted region (Supplementary Table 3). The virulence plasmids of C1789 and C4599 were almost identical to pK2044 (Supplementary Fig. 6). The region was knocked out in C4599, but it was reversed by two fragments (*hp1* to *hp4, hp5*) in C1789.

Among the differentially expressed genes (DEGs) caused by the knockout of the ~3 Kb region, the expression of 151 genes was upregulated with a fold change ranging from 1.26 to 11.44-fold and an adjusted *p* value < 0.05 (Supplementary Fig. 7c). A series of genes

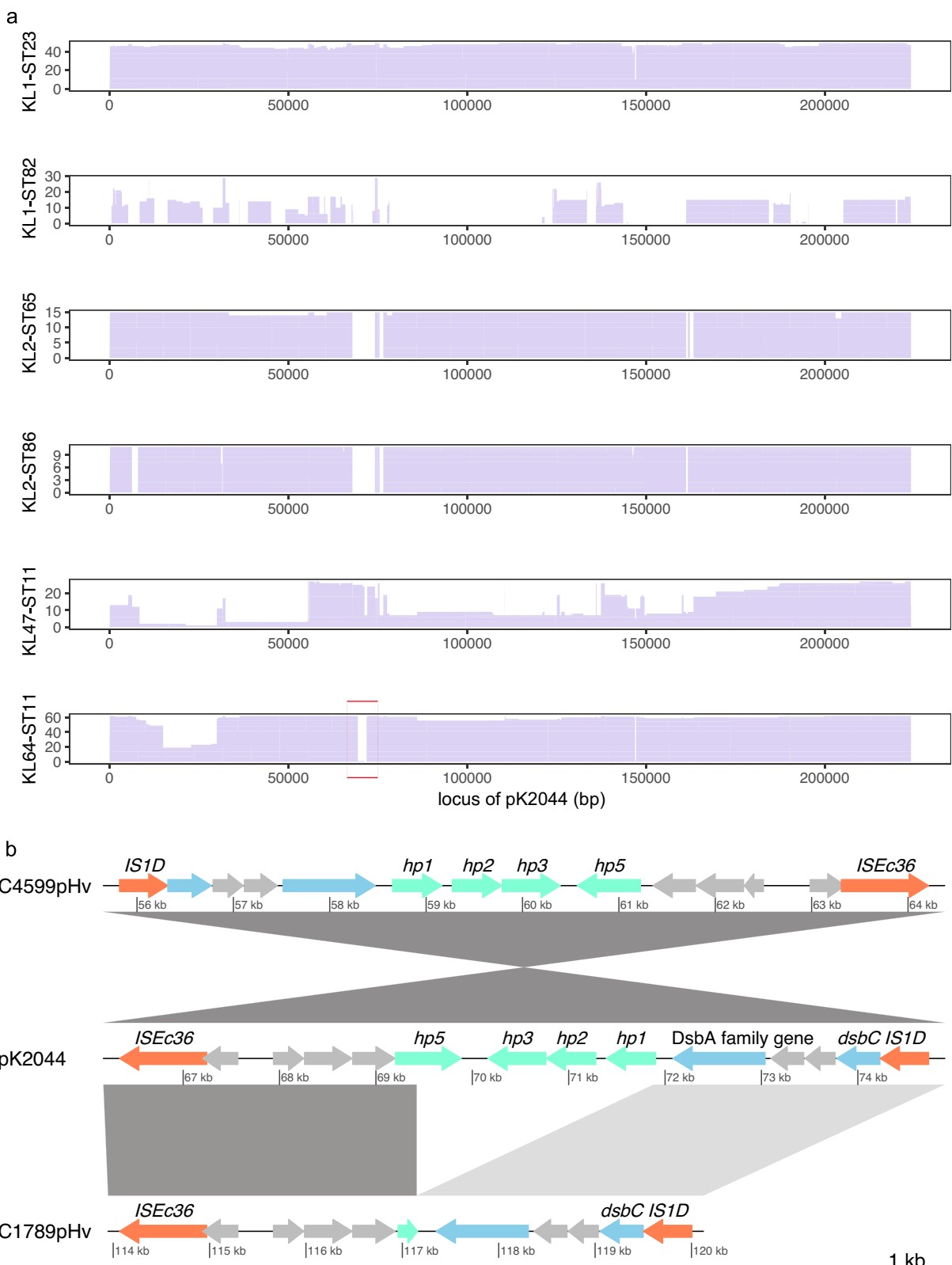

**Fig. 4 | The comparison of virulence plasmids of KL1, KL2, KL47, and KL64 *K. pneumoniae* with pK2044. a** Length distribution subset by serotype and MLST. The *x*-axis is alignment position (bp) with pK2044 as a reference. The *y*-axis is number of predominant serotypes and STs. The ~3 K deletion region on virulence plasmids of KL64 strains is marked by red box. **b** Linear comparison of ~3 K region in pK2044 and representative strains. Orange arrows: mobile elements; Green arrows: *hp1-3* and *hp5* in ~3 K region; Gray arrows: hypothetical genes. Blue arrows: genes with names or known products. pTJ12019Hv: a virulence plasmid of ST23-KL1 hvKP TJ12019; pC1789Hv: a virulence plasmid of ST11-KL64 hv-CRKP C1789; pC4599Hv: a virulence plasmid of ST23-KL1 CR-hvKP C4599. Visualized with genoPlotR v0.8.11[67]. Source data are provided as a Source Data file.

encoding catalytic enzymes involved in the methionine metabolism pathway were upregulated, including *metB* (fold change: 3.28, adjusted $p = 0.003$), *metF* (fold change: 2.15, adjusted $p = 0.034$) and *mtnK* (fold change: 3.94, adjusted $p = 2.39E-04$). Beside *hp3* and *hp5*, among the 36 DEGs were downregulated at least a half fold change. In plasmids, among the 51 genes exhibiting opposite trends in expression between knockout and reversed groups, 24 genes were annotated as hypothetical proteins (Supplementary Fig. 7a, b). The expression of *bla*$_{KPC-2}$ was upregulated 1.5–2 fold in ~3 Kb region reversed strains. However, there was no difference in susceptibility to carbapenem agents for these wildtypes, knockout and reversed strains, indicating this region is not involved in carbapenem resistance.

To determine further the functionality of this region, KEGG pathway and GO analysis were performed. Enrichment of KEGG pathway analysis indicated cysteine and methionine metabolism (rich factor: 0.48, adjusted $p = 6.05E-09$) was the most affected with 12 upregulated genes in C4599Δ3K (Fig. 5a). Catalytic enzymes encoded by these upregulated genes *metL* (fold change: 2.52, adjusted $p = 0.009$), *metA* (fold change: 2.47, adjusted $p = 0.025$), *metB*, *metC* (fold change: 2.21, adjusted $p = 0.017$) and *metE* (fold change: 6.41, adjusted $p = 0.094$) promoted L-methionine biosynthesis. The expression of HTH-type transcriptional regulator gene *metR* (fold change: 4.20, adjusted $p = 0.050$) also increased, which regulated transcription of *metE*. The S-adenosylmethionine synthase encoded by upregulated gene *metK* (fold change: 2.37, adjusted $p = 0.017$) catalyzed the conversion of L-methionine to S-adenosyl-methionine (SAM), while increased expression of *mtnK*, *mtnA* (fold change: 2.43, adjusted $p = 0.002$), *mtnC* (fold change: 2.60, adjusted $p = 0.001$), *mtnD* (fold change: 2.30, adjusted $p = 0.008$) facilitated L-methionine recycling from SAM (Fig. 6). In addition to increased endogenous methionine biosynthesis, the expression of *metN* (fold change: 2.72, adjusted $p = 0.003$), *metI* (fold change: 2.72, adjusted $p = 0.002$) and *metQ* (fold change: 2.23, adjusted $p = 0.025$) encoding an ATP-binding cassette (ABC) transporter composed of three subunits were upregulated, which was able to acquire D-methionine (D-Met) from extracellular source.

When *hp5* was reversed into C1789, genes associated with the methionine metabolism pathway (rich factor: 0.65, adjusted $p = 0.023$) were significantly downregulated (Fig. 6). Hp1–4 reversed led to genes enriched in the phosphotransferase system pathway (rich factor: 0.19, adjusted $p = 9.97E-06$) being significantly downregulated. This may be related to the presence of *hp3*, which was predicted to encode a phosphatase. Although the DEGs upon C1789 reversed are enriched in multiple pathways, the set of these exhibiting expression changes opposite to those observed upon C4599 knockout are predominantly concentrated in the methionine metabolism pathway (quadrants 2 and 4 in Supplementary Fig. 8a, b).

## The deletion region was associated with oxidative stress in the transcriptome

Knockout of the ~3 Kb region caused upregulated expression of 22 genes enriched in the oxidoreductase activity (Fig. 5b), for instance, the oxidative stress response genes *trxC* (fold change: 3.21, adjusted $p = 3.91E-06$) and *grxA* (fold change: 3.15, adjusted $p = 1.51E-4$). The hydrogen peroxide-inducible gene activator *oxyR* and OxyR-regulated genes *katE* (catalase HPII), *katG* (catalase-peroxidase), *trxC* (thioredoxin), *ahpC* (alkyl hydroperoxide reductase C), *dps* (stress response DNA binding protein) and *suf* operon (Fe-S assembly proteins) were upregulated after knockout of the ~3 Kb region (Supplementary Fig. 8c). Besides, encoding superoxide dismutase *sodABC* were upregulated, which could protect cells against oxidative stress. Although the expression of peptide methionine sulfoxide reductase genes *msrA* and *msrB* was not significantly upregulated after ~3 Kb knockout in C4599, their expression was significantly downregulated when the region was reversed in C1789. It is known that the antioxidant defense systems relied heavily on the regulated reactivities of cysteine and methionine or their derivates[15]. All the above therefore suggest that the ~3 Kb region may play a role in defense from oxidative stress.

## The deletion region partly reduced the antioxidant capacity of ST11 hv-CRKP

To obtain functional validation of the RNA-seq data, we performed several assays to assess the effect of ~3 Kb region on the antioxidant capacity of ST11 hv-CRKP (Fig. 7). The total antioxidant capacity of reversed ~3 Kb region strains was significantly lower than wildtype with empty vector by measuring ABTS [2,2′-azinobis-(3-ethylbenzothiazoline-6-sulfonic acid)] radical clearance after replenishment of *hp1–4* [(79.63 ± 2.63)% vs (71.62 ± 1.40)%, $p = 0.010$] and *hp5* [(79.6 ± 2.63)% vs (72.96 ± 1.70)%, $p = 0.021$]. Furthermore, superoxide dismutase (SOD) activity also decreased after replenishing *hp5* (1.55 ± 0.31 vs 0.50 ± 0.15, $p = 0.005$), which was an endogenous enzymatic antioxidant that converts superoxide ions to hydrogen peroxide in cells[16]. Within the stimulation of 1 mM $H_2O_2$, the intracellular reactive oxygen species (ROS) of C1789::hp1–4 (0.10 ± 0.00 vs 0.13 ± 0.01, $p = 0.016$) and C1789::hp5 (0.10 ± 0.00 vs 0.16 ± 0.11, $p = 0.002$) both increased significantly, which meant less enzymes to mitigate the ROS comparing with the wildtype. In similarly, the growth of C1789::hp1–4 ($p = 0.022$) was slower in 5 mM $H_2O_2$, while C1789::hp5 grew as fast as the wildtype strain.

The bacterial survival in macrophages was determined for C1789, C4599, and their mutants. After replenishment, the survival rates of C1789::hp1–4 [(12.81 ± 1.12)% vs (3.08 ± 0.28)%, $p = 1.30E-04$] and C1789::hp5 [(12.81 ± 1.12)% vs (4.53 ± 0.66)%, $p = 3.90E-04$] in macrophages decreased significantly (Fig. 7e). Under the laser scanning confocal microscopy, it was a clear indication that the reversed strains demonstrated a significantly reduced capacity to persist within the intracellular environment of macrophages compared to the wildtype strain (Fig. 7f, Supplementary Figs. 9 and 10). The results of C4599 and its mutants revealed a trend contrary to that observed in the ST11-KL64 (Supplementary Fig. 11a). The knockout of the deletion region led to an increased, albeit non-significant, survival rate, whereas the reversed *hp5* mutants exhibited a significant decrease in survival rates [(1.48 ± 0.31)% vs (0.31 ± 0.10)%, $p = 0.004$]. This observation suggested that the presence of the ~3 Kb region negatively impacts bacterial survival within highly oxidative environment of macrophages.

ABTS activities determination assays were also conducted on ten ST11-KL47 and ten ST11-KL64 hv-CRKP strains for comparing the antioxidant capacity of ST11 hv-CRKP with and without the pK2044-like plasmid (Supplementary Fig. 11b). The result showed that the antioxidant capacity of KL64 hv-CRKP was significantly higher than that of KL47 ($p = 0.002$), indicating this plasmid might contribute to the success of KL64 sub-lineage in ST11 hv-CRKP.

## Discussion

The clonal spread of CRKP has resulted in its global expansion and progression towards increased virulence, driven by several factors, including the mutual adaptation between host bacterial chromosomes and plasmids. Assessing the impact of ensuing genetic variations warrants further investigation. In this study, long-read sequencing techniques were employed to obtain a multitude of complete genomes, elucidating the evolutionary trajectories of chromosomes and virulence plasmids. These insights regarding the evolution of virulence plasmids could offer novel perspectives on the virulence mechanism of ST11 hv-CRKP and potentially shed light on why some strains carrying virulence genes fail to exhibit hypervirulence.

We compiled a dataset of ST11 *K. pneumoniae* to explore hallmarks of acquired virulence. Most of the included isolates were sampled in China due to our data collection and generation as well as ST11 being the dominant clone. A prior research[17] indicates subclone replacement occurring within the predominant clone ST11. The point

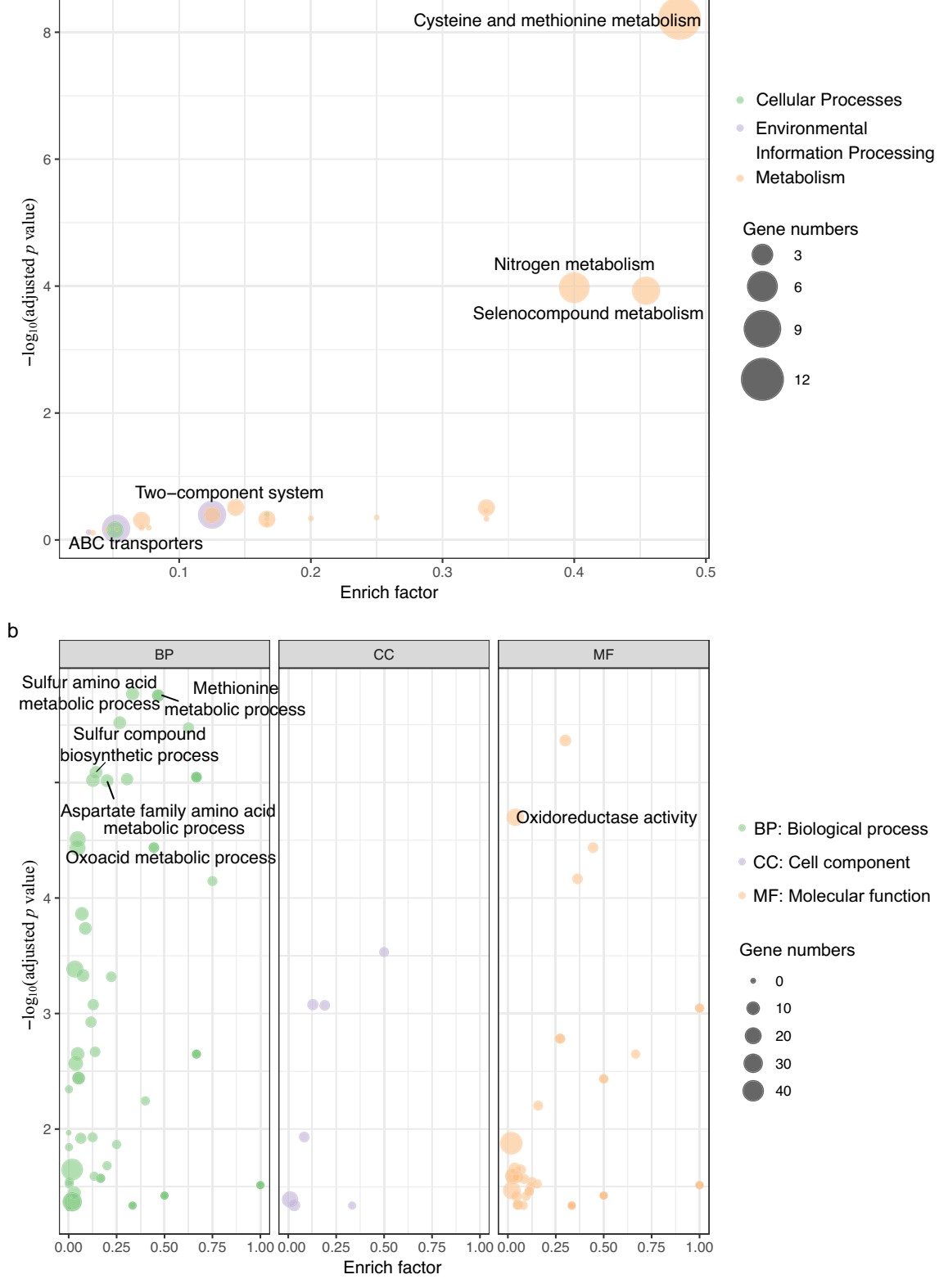

**Fig. 5 | Enriched KEGG pathway and GO analysis of differential expressed genes between wildtype and knockout strains based on their functions. a** Enriched KEGG pathway terms of all DEG. **b** Enriched GO terms of upregulated genes. The *x*-axis is the enrich factor, which is the ratio of DEGs numbers annotated in the pathway term to all gene numbers annotated in the pathway term. The *y*-axis is the minus $\log_{10}$ scale of the adjusted *p* values [$-\log_{10}$ (adjusted *p* value)], which indicates the significant level of expression difference. Fisher's exact test was used to test significance and adjusted by Benjamini and Hochberg. Source data are provided as a Source Data file.

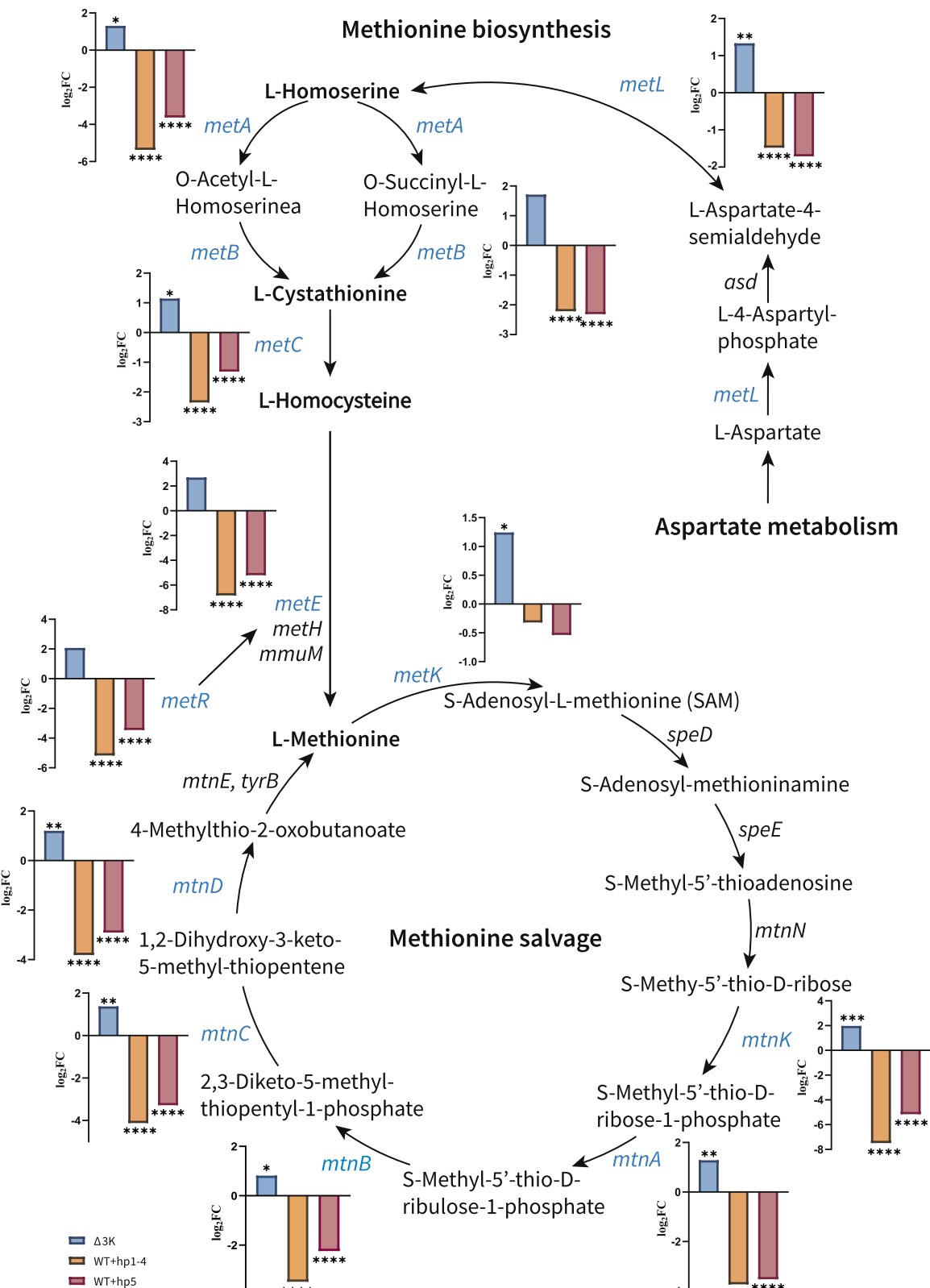

**Fig. 6 | Gene expression changes in methionine biosynthesis and salvage pathway influenced by the ~3 K region.** Catalytic enzyme genes are shown in *italic*. Blue italics represent genes with significantly different expression, and the adjacent bar graphs show the log$_2$ scale of the fold change (log$_2$FC) of differentially expressed genes (DEGs) after knockout or reversal based on RNA-seq. The blue bars (Δ3 K) represent the comparison between C4599 and C4599Δ3K, the yellow bars (WT+hp1-4) represent the comparison between C1789::hp1-4 and C1789pEasy, and the red bars (WT+hp5) represent the comparison between C1789::hp5 and C1789pEasy. Wald test adjusted for multiple testing using the procedure of Benjamini and Hochberg was used to test significance, and adjusted $p$ value < 0.05 was considered statistically significant when comparing the wildtype and its mutant. *adjusted $p$ < 0.05, **adjusted $p$ < 0.01, ****adjusted $p$ < 0.0001. Source data are provided as a Source Data file. Metabolic pathway reference: https://www.kegg.jp/entry/map00270.

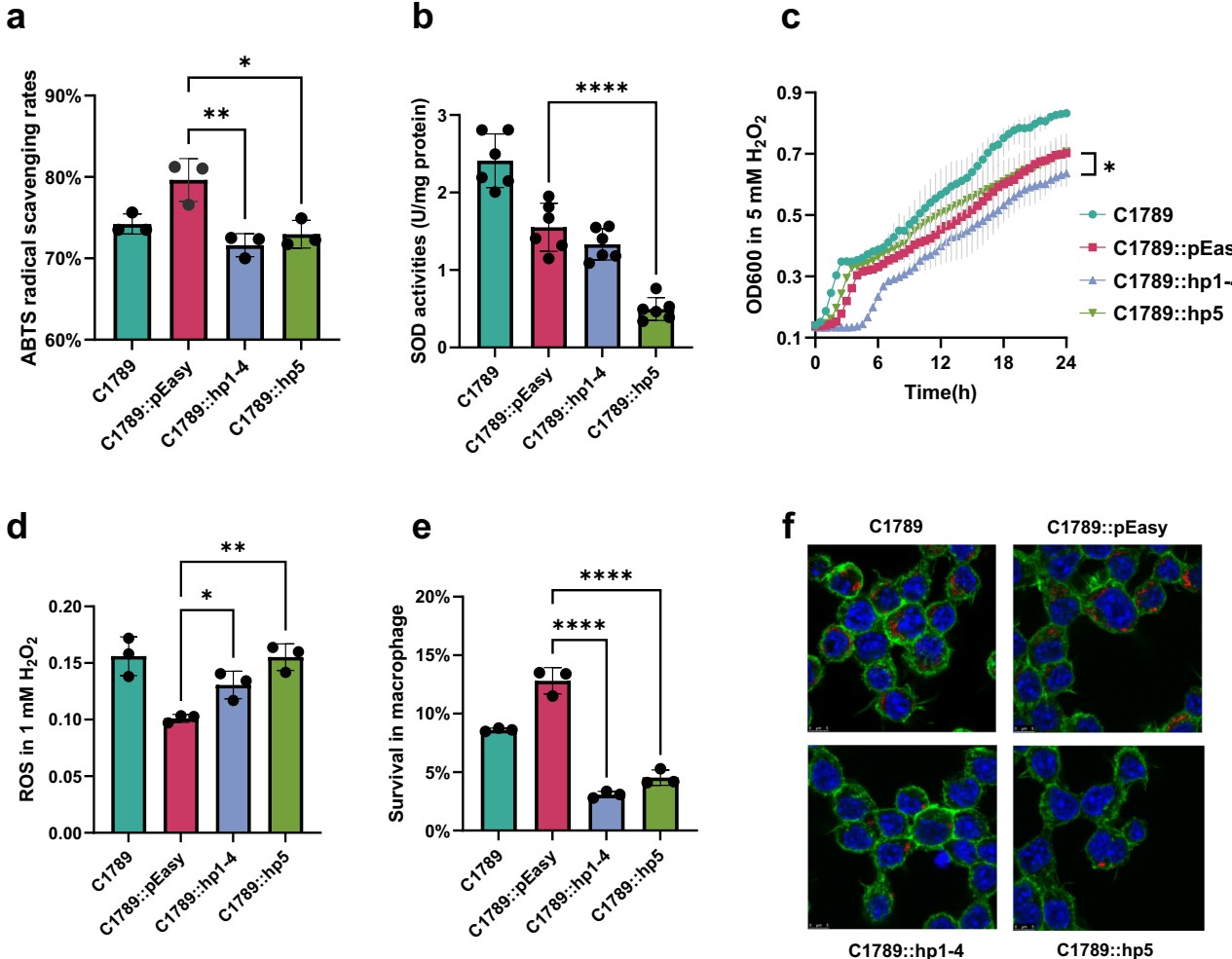

**Fig. 7 | Antioxidant capacity of wildtype and ~ 3K region reversed ST11 hv-CRKP.** Antioxidant capacity of ST11 hv-CRKP C1789 were assessed by **a** ABTS and **b** SOD activities determination assays. Relative scavenging rates was the calculated clearance per concentration unit. The SOD activity assay was performed with six biological replicates. \*$p = 0.021$, \*\*$p = 0.010$, \*\*\*\*$p = 2.01E\text{-}05$. **c** The growth curves of strains in 5 mM $H_2O_2$. Growth differences were determined by comparing average growth rates over a two-hour period during the logarithmic growth phase (\*$p = 0.02$). **d** The intracellular ROS of strains after stimulation with 1 mM $H_2O_2$. \*$p = 0.016$, \*\*$p = 0.002$. **e** The survival rates of C1789 and its reversed strains in

macrophage. \*\*\*\*$p < 0.001$. **f** Utilizing laser confocal microscopy to visualize the survival of bacteria within macrophages. F-actin were stained by rhodamine phal-loidin (green), and nuclei were counterstained with DAPI (blue). The bacteria were labeled with pHrodo (red). Scale bar = 5 μm. The representative result is present in **f**, and other images could be found in Supplementary Figs. 9 and 10. Two-tailed unpaired t-tests were used to compare C1789::pEasy *vs.* C1789::hp1-4/C1789::hp5 in **a**–**e**. The bars and error bars present mean with SD. Three biologically independent experiments were performed for assays in **a**, **c**–**f**. Source data are provided as a Source Data file.

mutation of *recC* identified in KL64 considerably facilitates recombination. A study hypothesized that KL64 strains evolved from KL47-like strains through recombination and caused significantly higher 30-day mortality than ST11-KL47 or non-ST11 CRKP[8]. However, this reconstructed phylogenetic tree could not distinguish whether they derived from a chromosome or plasmid. And the evolution of the two genetic materials does not coincide, with plasmids being more active in their horizontal transferring. In this study, the application of long- and short-read sequencing was conducive to assemble the complete genome to perform phylogenetic analysis on the chromosomes of ST11 CRKP and virulence plasmids, respectively. We discovered a potential connection between the plasmids and the chromosome, that was pK2044-like virulence plasmids originating from ST23-KL1 tend to be prevalent in a specific lineage of ST11-KL64. In China, it was reported that at least two virulence genes have been identified in clinical isolates of ST11 CRKP in the past decades[18,19]. HvKP may emerge as a significant threat to human health in the future[20]. Genomic analysis of global *K. pneumoniae* plasmids indicates that, in the majority of cases, the formation of hv-CRKP is a result of the acquisition of virulence genes by

classical CRKP strains[21]. The acquisition of pK2044-like plasmids by ST11-KL64 has been associated with an increase in virulence and a higher mortality rate in clinical infections[22]. However, key virulence loci encoding aerobactin (*iuc*) and salmochelin siderophore (*iro*) are at a high frequency within the ST23 lineage[23].

The different serotypes of ST11 strains carried a variety of virulence plasmids. Despite ready mobilization and the frequent exchange of genetic material, the virulence plasmids identified in ST11-KL64 strains showed high genetic similarity, suggesting a single, fairly recent origin. ST11-KL47 hv-CRKP frequently lacked the typical virulence marker genes *rmpA/rmpA2* and siderophore gene clusters[9]. In this work, we selected strains which carried the four virulence genes for sequencing, resulting in a higher representation of ST11-KL64 compared to ST11-KL47 *K. pneumoniae* strains.

In ST11-KL64, a deletion within the virulence plasmids was identified, which could represent an accidental event that has been preserved and inherited in ST11 hv-CRKP. In this region, *hp5* appears to be crucial, as its predicted protein structure matched that of the *Escherichia coli* protein YaaA. YaaA is a key element of the long-term stress

defense against ROS[24]. We suppose that Hp5 serves a similar function, but unlike YaaA, does not act to defend against oxidative stress. ROS, SOD activity and ABTS radical scavenging assays confirmed that it was connected to the antioxidant capacity of hv-CRKP. Hp5 might have a multifaceted effect to maintain intracellular redox homeostasis, rather than simply acting on a single pathway. Since the later part of Hp5 was lost in ST11 hv-CRKP, it was unclear whether the incomplete protein could perform similar or new functions. In addition, the study found that Hp1–4 may also be involved in antioxidation and Hp3 was probably a reductase in the respiratory chain. Thus, we supposed that hv-CRKP tended to lose *hp5* to improve its antioxidant capacity, and *hp1–4* may also therefore be involved in this antioxidant process.

Altered metabolism of methionine may provide an explanation for how the ~3 Kb region affects the antioxidant capacity of hv-CRKP. Methionine is a well-known amino acid that contains sulfur and is necessary for the initiation of protein synthesis. Due to the electron-rich sulfur atom in the side chain, methionine and cysteine are more vulnerable to oxidation and converted into methionine sulfoxide and sulfonic acid, respectively[25]. Oxidized cysteine residues are reduced by thioredoxins and glutaredoxins under physiological or oxidizing conditions[25]. Methionine sulfoxide is reduced back to methionine by methionine sulfoxide reductases (Msr) for repairing oxidatively damage. The oxidized Msr, thioredoxin, and thioredoxin reductase are in turn reduced by the latter one[26]. Thus, it was believed that methionine residues in proteins can provide antioxidant defense through the recycling pathway[27]. In our study, the ~3 Kb region deletion increased the expression of several oxidoreductases; MsrA and MsrB decreased both free methionine sulfoxide and methionine sulfoxide in proteins to repair oxidized methionine[15]. When the ~3 Kb region was reversed, the expression of these *msr* genes was significantly reduced. We are therefore confident that the ~3 Kb region plays a role in the synthesis of methionine and cysteine as well as the anti-oxidation of their metabolites to prevent oxidative damage.

The professional phagocytes serve as the immune system's first line of defense against infections and invading pathogens. Upon macrophage activation, there is a dramatic increase in ROS production during cellular stress. Phagocytes primarily generate toxic ROS through phagosomal NADPH oxidase-dependent respiratory bursts[28]. In addition to NADPH oxidase, mitochondrial ROS (mROS) are produced via the mitochondrial oxidative phosphorylation mechanism[29]. Subsequently, we noted that the survival rate of ST11-KL64 hv-CRKP within macrophages diminished following complementation in the ~3 Kb region, potentially because bacteria became more susceptible to oxidative damage.

In conclusion, the increasing prevalence of ST11 hv-CRKP poses a significant threat to public health with mobility of virulence associated plasmids an additional concern. We observed that the classic virulence plasmid pK2044-like plasmid in a ST11-KL64 lineage lost a ~3 Kb region. The lost fragment of pK2044-like plasmids likely acts to enhance the antioxidant capacity of ST11-KL64 hv-CRKP. Therefore, this study provides important insights into the evolution of virulence plasmids in ST11 CRKP and sets the foundation for preventing the survival of these threatening bacterial pathogens.

## Methods

### Definitions and dataset compilation
Given the longstanding controversy regarding the virulence, *K. pneumoniae* in this study was classified into four distinct groups based on the presence or absence of carbapenemase and virulence factors analyzed by the Kleborate v0.3.0[30] software (Supplementary Table 4).

We compiled a whole-genome database of 637 *K. pneumoniae* isolates sampled from 84 hospitals in 25 provinces and municipalities across China between 2005 and 2019. The strains collected in this study were derived from several long-term Chinese multicenter surveillance conducted by our team, as well as strains collected in

previous studies[2,4,31–37]. Most of the isolates were from the CRE network and the Chinese Antimicrobial Resistance Surveillance of Nosocomial Infections (CARES) project, which collected strains from hospitals in various provinces and cities in China for antimicrobial susceptibility testing and sequencing of resistant or virulent strains. The sampled methods have been described in the previous articles. Based on these efforts, a sequencing database containing 637 *K. pneumoniae* has been established. A hybrid sequencing approach was applied to 302 *K. pneumoniae* isolates selected as representative according to their collection date and specimen source. This study was approved by Peking University People's Hospital Institutional Review Board (No. 2019PHB194-01).

We also downloaded *K. pneumoniae* assemblies, along with their associated metadata, from the NCBI Reference Sequence database (as of 16th June 2020) to investigate global patterns of CRKP and contextualize our data. We set out to identify, from this dataset, sufficiently contiguous assemblies for the resolution of bacterial to chromosome and plasmid. The chromosomal genome of *K. pneumoniae* is usually around 5 Mb in size, with several plasmids. Therefore, criteria were devised to select publicly available assemblies long enough to be studied, including (1) longest contig length > 4.5 Mb, and (2) number of contigs <40. Following filtering according to those criteria, a total of 1219 assemblies were considered from the NCBI Reference Sequence database (*n* = 917) and our database (*n* = 302).

### Genome sequencing and annotation
Genomic DNA of single clones at an exponential phase was extracted using a TIANamp Bacteria DNA Kit (TianGen, China). DNA was sent to Tianjin BioChip (Tianjin, China) for library preparation and sequencing. The short-insert (~500 bp) paired-end libraries were constructed and sequenced on the Illumina NovaSeq platform (Illumina, USA). The raw reads were subsequently trimmed for quality using Trimmomatic v0.35[38] and fastp v0.20.1[39] with default parameters. For long-read sequencing, the genomic DNA was sheared to ~50 Kb fragments, of which smaller than 7 Kb fragments were filtered out. SMRTbell library was constructed and sequenced on the PacBio Sequel II system (Menlo Park, CA, USA). Hybrid assembly was conducted in Unicycler v0.4.7[40] using the 'normal' mode. Genomes and full metadata utilized for analysis are available on NBCI under the BioProject: PRJNA1015184.

Prokka v1.13.7[41] was applied to annotate all assemblies (*n* = 1219). Sequence type (ST), serotypes, virulence genes, virulence score, and antimicrobial resistance genes were assigned using Kleborate v0.3.0[30] and Kaptive v2.0.0[42]. Protein structure was predicted using Phyre2 v2.0[43], powered by ColabFold v1.3.0[44], and then searched in the AlphaFold protein structure database[45] using Foldseek (search.foldseek.com)[46] with default parameters.

The location of major virulence associated genes, including *rmpA*, *rmpA2*, *iroN*, *iucA*, and *pagO* (also named *peg-344*) were extracted sequences from pLVPK (AY378100.1) and pK2044 (NC_006625.1). The BLASTn[47] v2.9.0+ analysis showed that most virulence genes were complete (at a nucleotide identity > 97% and coverage over the gene > 90%), while over 20% of the sequences only aligned with the first 50% of *iroN*. The incomplete alignment of *iroN* was marked with *iroN**. The contigs carrying these virulence genes were considered virulence contigs and used for subsequent evolutionary analysis. PlasmidFinder v2.1[48] was applied to identify known plasmid replicons.

### Phylogenetic analysis and comparison
Phylogenetic analysis was conducted for 246 assemblies that belonged to the ST11 clones (predominately serotypes KL47 and KL64) after removing a length of longest contig < 4.5 Mb and > 40 total contigs in the assembly. The phylogeny was built using RaxML v8.2.12[49] specifying the GTRGAMMA model with 1000 bootstrap on a 758,264 bp core-genome sequence alignment, which was filtered to remove recombinant regions (55,262 bp) identified by ClonalFrameML v1.11[50] with 100

pseudo-bootstrap replicates. SNP-sites 2.5.1[51] was used to extract SNPs and create the final multiple sequence alignment (MSA). ZJ08067 was designated as the root due to its earliest collection date, which occurred in April 2008. TempEst v1.5.3[52] was employed for preliminary assessment of temporal signal, while BETS[53,54] were utilized for formal evaluation with an initial Markov chain of 10 million and 50 path steps with a chain length of 500,000 iterations for each power posterior. A rooted, time-measured phylogeny was constructed using BEAST2 v2.7.4[55]. BEAST2 was employed to execute chains of length 100,000,000, with a 10% burn-in, logging every 10,000 iterations. Prior assumptions of a coalescent constant model and a strict clock rate were used. Convergence of the Markov chain Monte Carlo (MCMC) chain was assessed using Tracer v1.7.2. The maximum clade credibility (MCC) tree for each model was generated using TreeAnnotator v2.7.4[56] and visualized using ggtree[57].

We additionally conducted an analysis focused on virulence associated contigs. We utilized fastANI v1.33[58] to assess the Average Nucleotide Identity (ANI) between virulence contigs pairwise. High diversity challenged the creation of a sufficiently long multi-sequence alignment to include all isolates. As a result, we focused on plasmids with high similarity (an ANI > 95 and coverage of > 80%) to the classic virulence plasmid pK2044 (NC_006625.1). Snippy v4.6.0 (https://github.com/tseemann/snippy) was used for mapping virulence contigs to pK2044 and obtaining the alignment to reconstruct a pK2044-like plasmid phylogeny. Serotypes, as a phenotypic characteristic, was employed to investigate whether different serotypes exhibit genetic variations in virulence contigs. The analysis was conducted using Scoary v1.6.16[59] with employing default parameters. Subsequently, the identified specific genes were reconfirmed using BLASTn[47] requiring a nucleotide identity above 95%, while also considering the coverage. Genes with low coverage were represented in lighter shades in Fig. 3. The final plasmid phylogenetic tree was visualized in R package ggtree v3.8.2[57].

### Genome editing assay

A ~ 3 Kb region of the ST23 hvKP virulence plasmid containing five hypothetical genes (named hp1 to hp5) was identified as consistently absent, following a BLASTn[47] analysis returning no alignments, when the plasmid was associated with ST11 CRKP across our datasets. To determine the function of this genetic region, C4599 was selected as representative of ST23 CR-hvKP due to their highest ANI to pK2044. The sequence of interest was knocked out using the clustered regularly interspaced short palindromic repeat (CRISPR)-Cas9 genome cleavage system based plasmids, pCasKP-pSGKP[60].

The gene reversing assay was used to ligate the target genes of hp1-4 and hp5, respectively with a modified pEasy-T1 vector (TransGen Biotech, China) by Gibson assembly and transfer to a ST11 KPC-2 strain C1789 (hv-CRKP). The strains and plasmid vectors used in this study are listed in Supplementary Tables 3 and 5. The primers used in the study can be found in Source Data.

### Transcriptome sequencing and analysis

RNA sequencing was performed with three replicates to quantify patterns of gene expression in isolates C4599, C4599Δ3K, C1789::pEASY, C1789::hp1-4, and C1789::hp5 to better understand the role of the ~3 Kb region. Both C4599 and C1789 harbor pK2044-like virulence plasmids as well as the classic KPC-2 plasmid, thereby mitigating the potential impacts from other plasmids. Total RNA of single clones at an exponential phase was extracted (total RNA > 2 μg) with three replicates and 150 bp paired end sequenced on the Illumina Novaseq platform. The Majorbio Cloud Platform (www.majorbio.com)[61] were used to analyze. SeqPrep v1.2 (https://github.com/jstjohn/SeqPrep) and Sickle v1.33 (https://github.com/najoshi/sickle) with the default settings were used to filter raw data. High quality reads in each sample were aligned to the reference assemblies of C1789 and C4599 using Bowtie 2 v2.5.2[62]

employing default parameters. Quantitative analysis of gene expression was conducted using RSEM v1.3.3[63] and standardized by TPM (transcripts per million). Differential gene expression analysis among samples was performed using DESeq2 v3.18[64] package. Gene Ontology (GO) and Kyoto Encyclopedia of Genes and Genomes (KEGG) pathway analysis were used to identify enriched functions and pathways of differential expression genes. Goatools v1.3.11[65] and KOBAS 2.0[66] are used to identify statistically significantly enriched GO term and pathway using Fisher's exact test. The purpose of performing false discovery rates (FDR) correction is to reduce the Type-1 error by BH (Benjamini and Hochberg). After multiple testing correction, GO terms and pathway with adjusted $p$ value ≤ 0.05 are significantly enriched in DEGs.

### Growth rate measurement

Growth curves of C1789 and the reversed strains were determined in the presence or absence of $H_2O_2$. The bacteria in the logarithmic growth stage were adjusted to OD600 of 0.01 and inoculated in 200 μL of LB broth with 5 mM final concentration of $H_2O_2$. The absorbance was measured in $OD_{600}$ every 30 min by microplate reader (Thermo Scientific, Germany). All assays were performed independently three replicates.

### Measurement of antioxidant capacity in vitro

In vitro antioxidant activity was assessed using the ABTS free radical scavenging capacity detections and SOD enzyme activity in C1789 and the reversed strains. The ABTS assay was also performed on ten samples each of ST11-KL47 and ST11-KL64 hv-CRKP, with the aim of assessing the disparity in antioxidant capabilities between these strains. The strains in logarithmic growth stage were used for ultrasonic extraction of total protein, and the protein concentration was measured using the enhanced BCA protein assay kit (Beyotime, China). The assays were performed with commercial kits (Solarbio, China) on 96-well plates and compared by two-tailed unpaired t-test.

As for intracellular ROS detections, strains were incubated overnight, then stained with 2',7'-dicholorfluorescein diacetate (DCFH-DA) at 37 °C for 20 min in dark as commercial kits described (Beyotime, China). The free DCFH-DA was washed away, and the cultures were stimulated with 1 mM $H_2O_2$ for 40 min. The evaluation of ROS was further measured by microplate reader (Thermo Scientific, Germany).

### Survival in macrophage assay

Survival in macrophage assays was performed to assess the antioxidant capacity of bacteria within the host organism. RAW264.7 murine macrophages were preactivated prior to being infected at a multiplicity of infection (MOI) of 100. The infected cells were incubated at 37 °C with 5% $CO_2$ for 2 h, followed by the addition of 300 μg/mL hygromycin to eliminate extracellular bacteria. After 30 min of incubation, cells were lysed with 0.2% Triton X-100 and diluted on LB agar plates to count CFUs. Three biological replicates were carried out per strain. The cell line was obtained from the Cell Resource Center, Peking Union Medical College (PCRC), and the most recent authentication was conducted on March 26, 2019.

A laser scanning confocal microscopy approach was utilized to observe the interaction between bacteria and macrophages directly. RAW264.7 were seeded onto glass coverslips in 24-well culture plates to form monolayers. Bacteria in the logarithmic growth stage were resuspended to pre-incubated with pHrodo red (Thermo Scientific, Germany) for 1 h. The bacteria and cells were cocultured with MOI of 100 at 37 °C with 5% $CO_2$ for 2 h. After inhibiting extracellular bacteria by hygromycin, the glass coverslips were fixed with 4% paraformaldehyde for 15 min, and permeate with 0.25% Triton X-100 for 15 min. Subsequently, coverslips were blocked with 1% BSA for 1 h. F-actin of cells were stained using Actin-Tracker Green-488 (Beyotime, China), while cellular nuclei by DAPI (Beyotime, China) for a Leica TCS-

SP8 (Wetzlar, Germany) confocal microscopy observation. pHrodo Red, which was excited at 560 nm and emitted at 587 nm; Actin-Tracker Green-488, which was excited at 495 nm and emitted at 518 nm; and DAPI, which was excited at 364 nm and emitted at 454 nm. The image acquisition was performed using the Leica Application Suite X 2.0.0.14332 software and the resolution was set to 1024 × 1024 pixels.

## Statistics and reproducibility

No statistical method was used to predetermine the sample size. In all assays, except for the SOD activity assay, each sample was subjected to three biological replicates. The SOD activity assay was performed with six biological replicates. The assay utilizing laser confocal microscopy to visualize the survival of bacteria within macrophages was also with three biological replicates but showed the representative result in Fig. 7. Other images can be found in Supplementary Figs. 9 and 10.

## Reporting summary

Further information on research design is available in the Nature Portfolio Reporting Summary linked to this article.

## Data availability

KEGG (https://www.kegg.jp/) and GO (https://www.geneontology.org/) database are used in this study. All new assembly data used for phylogenetic analysis have been submitted to in GenBank and assigned the BioProject accession number PRJNA1015184. Accession numbers and the processed data generated in this study are provided in the Source Data. Some new assembly data of the total genome dataset (Supplementary Fig. 1 showed) are not publicly available for other unfinished projects but are available from the corresponding author on reasonable request. The metadata can be found in Source Data. Transcriptome data of knockout and reversed strains has been submitted to the Sequence Read Archive (SRA) and assigned the BioProject accession number PRJNA1047885. Phyre2 (http://www.sbg.bio.ic.ac.uk/phyre2) and AlphaFold protein structure database (https://alphafold.com) were used for protein structure prediction. Source data are provided in this paper.

## Code availability

No custom code or mathematical algorithm was utilized in this study. All analyses were conducted using open-source software, which has been explicitly stated and referenced in the method section.

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

## Acknowledgements

We thank Dr. Hua Gao for his valuable guidance and advice on bioinformatics analysis in the study. We also thank Dr. Chu Wang and his team from the College of Chemistry and Molecular Engineering, Peking University for their help in protein structure prediction. This study was partly supported by the National Natural Science Foundation of China under award numbers 81991533 (to H.W.) and 82102397 (to R.W.).

## Author contributions

H.W. and R.W. conceived, designed, and supervised the study. R.W., Q.W., Y.Z., X.Wa., L.J., and A.Z. collected bacterial isolates. R.W., A.Z., S.S., G.Y., Q.D., X.Wu, F.C., and Y.Z. performed phenotypic tests and gene editing assays. Bioinformatic analyses were performed by R.W. and S.W., L.v.D. and F.B. supervised it. R.W. wrote the draft, H.W., L.v.D., and F.B. revised it. All authors read and approved the final version of the manuscript. The corresponding author attests that all listed authors meet the authorship criteria and that no others meeting the criteria have been omitted.

## Competing interests

The authors declare no competing interests.
