## [Peer Review File · Nature Communications]

REVIEWER COMMENTS

Reviewer #1 (Remarks to the Author):

This is a solid study of the evolutionary of a *K. pneumoniae* subclone of ST11-KL64. The authors collected an impressive genome data set from online data bases and also data generated here. They convincingly demonstrate that this subclone acquired a virulence plasmid from from KL1 and that it is characterised by a major deletion.

I don't have major concerns with this work, but rather a few important suggestions to try to harness the power of such genomic data set.

- Please do not report p-values for root-to-tip regressions. The typical program used for this purpose, TempEst, does not report them because they are statistically invalid, due to pseudoreplication. Instead, just reporting the R^2 value would suffice. I also strongly suggest showing the regression. There are some *K. pneumoniae* data sets that have sufficient temporal signal and sometimes this is obvious from the regression.

- The root-to-tip regression can sometimes be misleading. I suggest conducting a formal approach to assess temporal signal, BETS (Duchene et al. 2020; see also https://beast.community/bets_tutorial) that has much higher sensitivity. This test can be effectively conducted under a simple coalescent model and I recommend setting a small prior on the population size (for example a gamma prior distribution (shape = 0.001; scale =

1,000.0)). If this analyses detects temporal signal, then it would be very interesting to tease out the evolutionary timescale of these lineages. I do understand that this entails additional analyses, so I leave it up to the authors whether they want to include this point.

- Why were samples with no collection times excluded, particularly given that they were not used to calibrate the molecular clock? Standard phylogenetic analyses (i.e. not using a molecular clock), are agnostic about the timescale. Moreover, if it is only a couple of samples, their age can be estimated under a Bayesian approach (https://beast.community/tip_date_sampling).

- Finally, it is very common practice to exclude recombining regions from bacterial genome data sets. However, it is important to report what proportion of the alignment was removed due to recombination. For example, what was the longest stretch removed from a sequence, or what was the average stretch removed across the alignment. I find this very useful to understand how much information could have been lost in recombination removal.

Duchene, S., Lemey, P., Stadler, T., Ho, S. Y., Duchene, D. A., Dhanasekaran, V., & Baele, G. (2020). Bayesian evaluation of temporal signal in measurably evolving populations. *Molecular Biology and Evolution*, 37(11), 3363-3379.

Reviewer #2 (Remarks to the Author):

The manuscript by Ruobing Wang and colleagues describes a functional genomics analysis of carbapenem-resistant and potentially hypervirulent *Klebsiella pneumoniae* (Kp). It is now well documented that the dominant carbapenem-resistant Kp lineage disseminated in China is sequence type 11 carrying the KPC-2 carbapenemase. Initially, dissemination of this clone was associated with the KL47 capsule synthesis locus but it is now clear that a recombinant derivative carrying the KL64 locus has displaced the KL47 progenitor. Worryingly, from a public health perspective, these multi-drug-resistant ST11-KL64 strains have also acquired a virulence plasmid carrying the key genes associated with hypervirulence, and which has been shown to increase virulence in mouse/*Galleria* models and increase patient mortality. Hence any works seeking to better understand the evolution and factors contributing to the widespread dissemination of these Kp are of high importance to the *Klebsiella* research and public health research communities.

Here the authors use genomic comparisons to compare the virulence plasmid acquired by ST11-KL64 to those carried by other ST11 and unrelated Kp e.g. ST23. They show i) that the ST11-KL64 plasmid is closely related to that carried by ST23, and ii) that ST11-KL64 strains are associated with a conserved deletion of ~3Kb in the virulence plasmid. Subsequently, RNASeq is used to explore the impact of the 3Kbp region on gene expression. The authors identify methionine metabolism and oxidative stress pathway genes enriched among those that are differentially expressed between ST11 wild type strains and their isogenic mutants with plasmid deletion reversal, as well as among ST23 wild type strains (with intact virulence plasmid) vs their isogenic plasmid deletion mutants. Finally, the authors demonstrate that the ST11 plasmid deletion reversal mutant has reduced antioxidant capacity and reduced macrophage survival compared to the isogenic wild type strain. The study presents a novel biological hypothesis for the rapid expansion of carbapenem-resistant and virulent ST11 Kp carrying the KL64 capsule locus- which will likely be of broad interest to the *Klebsiella* research community. While the general approach as described is sound, there are many missing methodological details – particularly regarding the RNASeq analysis (see below). Additionally, the results section is drastically lacking in quantitative description (see below examples) and the underlying data are not available (neither sequence data nor processed RNASeq analysis data) so it is not possible to fully assess the quality of this evidence. Additionally, the conclusion that the 3Kb deletion has contributed to the success of ST11-KL64 (e.g. as opposed to ST11-KL47) is somewhat circumstantial at this stage because there is no direct comparison of RNASeq nor oxidative stress / macrophage survival between ST11-KL64 with /without the deletion and ST11-KL47.

Finally, regarding the findings about the evolutionary origins of the ST11-KL64 virulence plasmid e.g. as stated at lines 418-423: “We discovered a potential link between the plasmids and the chromosome and hypothesized that pK2044-like virulence plasmids transfer into ST11- KL64 CRKP from ST23-KL1, even though it was challenging to determine when the virulence plasmid movement happened. In a word, virulence plasmids from ST23-KL1 were acquired and independently evolved in ST11-KL64, and then spread for the past decades.” I agree this is very likely the case and is supported by the data presented here. However, this is also something that I would expect given the data that have already been reported in the literature about the similarity between ST11-KL64 virulence plasmids and pK2044 (e.g. Ruan et al *Infect Drug Resist.* 2020; 13: 199–206. Guo et al *Front Microbiol.* 2022; 13: 929826. etc), as well as what is known about the diversity of virulence genes / plasmids generally (see Lam et al *Genome Med* 2018 Oct 29;10(1):77. doi: 10.1186/s13073-018-0587-5, Spader et al. *Genome Med* 2023 Jan 19;15(1):3.doi: 10.1186/s13073-023-01153-y.) and the dominance of ST23 among hvKp, including in China (e.g. see Liu et al *Virulence* 2020 Dec;11(1):1215-1224.). It would be pertinent to include a discussion of the existing literature to contextualise the findings described here and tone down the originality claims.

MISSING METHODOLOGICAL DETAILS

1. Please describe the isolate collection from which the isolates in the current study were derived. There is a description of an ethics approval but no description of the study or collection.
2. DNA extraction and sequence library preparation methods should be provided for the new sequence data generate here. For the Illumina data please also indicate if single or paired end and state the read length.
3. RNA extraction and library prep and sequencing details (e.g. read lengths) should be stated. The approach for processing these data should also be stated – presumably some sort of referenced based approach was used? If so, which mapping algorithm, parameters, what reference sequences were used?
4. Please indicate if any read filtering or QC steps were undertaken prior to genome assembly and on the assemblies themselves (including those downloaded from NCBI).
5. Genome accessions should be provided. Both the project IDs and the individual isolate / experiment accessions must be provided for the purposes of reproducibility.
6. Lines 85-90: “that were sufficiently contiguous to allow resolution of chromosome and plasmid, to investigate global patterns of CRKP and contextualize our data. We set out to identify, from this dataset, sufficiently contiguous assemblies for resolution of bacterial chromosome and plasmid. Following

filtering according to those criteria, a total of 1,219 assemblies were considered from the NCBI Reference Sequence database (n = 917) and our database (n 90 = 302).” The specific criteria should be stated.

7. Lines 97-99: “Please indicate any parameters or thresholds used for structural analyses and to filter UniProt or FouldSeek outputs.

8. Line 101: Please indicate which type of BLAST (e.g. BLASTn, tBLASTn) and thresholds used to confirm virulence gene locations.

9. Lines 102-104: “We selected 103 virulence contigs from 201 isolates in the predominant serotypes KL1, KL2, KL47, or KL64 to analyze.” What criteria were used to select these sequences?

10. Line 110-111: Please specify the parameters that were used to run Roary and the definition of core i.e. $\geq 95\%$, 100% etc. Also state any parameters for ClonalFrameML. What was the length of the sequence alignment used for ClonalFrame analysis and how much sequence was masked by this analysis?

11. Lines 115-116: “we focused on plasmids with high sequence identity (a coverage of $> 95\%$) to the classic virulence plasmid pK2044 (NC_006625.1)” Should this indicate that you focussed on plasmids with high sequence coverage as specified within the parenthesis? Or if both identity and coverage cut-offs were applied, please indicate both here.

12. Lines 118-120: “Correlation between genes annotated on identified virulence plasmids and discrete phenotypic characteristics were analyzed using Scoary and verified using BLAST.” Which phenotypic characteristics were tested and how were these defined? Please also state any parameters for Scoary and indicate BLAST type/thresholds.

13. Lines 120-121:” The final plasmid phylogenetic tree was visualized in R. – Please name and cite the appropriate R packages.

14. Lines 125-126: “was identified as consistently absent when the plasmid was associated with ST11 CRKP across our datasets.” How was absence determined?

15. Lines 137-139: "RNA sequencing was performed with three replicates to quantify patterns of gene 138 expression in isolates C4599, C4599Δ3K, C1789::pEASY, C1789::hp1-4, and 139 C1789::hp5 to better understand the role of the ~3 Kb region." Why were these strains chosen and not others? E.g. Why wasn't TJ12019 and its mutant derivatives as described in lines 126-130 used?

16. Line 143: "The Majorbio Cloud Platform (www.majorbio.com) and R were used to analyze the data." Please state and cite the R package(s) used.

17. Lines 201-202: "however, none of these isolates harbored virulence plasmids based on the presence of virulence associated marker genes." Which marker genes were used?

18. Lines 218-219: "However, because of the weak temporal signal in the data ($R^2 = 0.12$, $P = 0.0001$)," How was temporal signal determined?

19. Line 222: "How was 'closely related' defined?"

LACK OF QUANTITATIVE DESCRIPTION

1. Line 225-227: "The virulence plasmid in KL64 strains were more similar to the classic virulence plasmid pK2044, while KL47 strains carried fewer virulence genes integrated into the chromosome." How was similarity to the pK2044 plasmid defined here? In terms of total number of virulence genes carried on the plasmid?

2. Line 307-308: Discussion of differentially expressed genes. Please describe a summary of the results e.g. total number of genes up / downregulated, and fold change ranges etc, and give specific fold change / p values for the genes discussed. These details are provided in the supplementary figure but given the relevance to the key findings it is important to state clearly here as well.

3. Lines 321-327: Do these lines refer to chromosomally encoded genes? Where are these data shown? Figure S4 appears to show only plasmid-encoded genes. Again, please give the specific details about the fold changes and p-values associated with these genes so that readers and reviewers can interpret and assess the findings.

4. Lines 329-330: "Enrichment of both analyses indicated cysteine and methionine metabolism to be the most affected with 12 upregulated genes in C4599Δ3K (Figure 6A)." Please explain and provide quantifiable evidence to support this statement, In Fig 6A (KEGG enrichment analysis) I see that the

point labelled for cysteine and methionine metabolism has the highest log fold change and highest enrich factor so this make sense. However, in 6B (GO analysis) I don't see any points labelled for cysteine metabolism and the methionine point has neither the highest fold change nor the highest enrich factor. Is there some reason that the other points should be discarded? Also please provide references for the information about the methionine pathway that is provided and shown in Figure S5.

5. Lines 341-353: - Discussion of up/down regulated genes in different mutants is missing quantitative evidence in the text and the figure shows no indication of a consideration of p-values or other appropriate approaches to exclude experimental noise.

OTHER MINOR COMMENTS

1. Line 31-33: "The pK2044-like virulence plasmid provides ST11-KL64 with evolutionary advantages through altering its genetic material to enhance survival." I don't see how this statement is supported by the data presented in this work. Suggest expanding or rephrasing,

2. Lines 42-45: "Over the past decade, hv-CRKP has rapidly disseminated, a phenomenon typically attributed to the acquisition 44 of virulence plasmids containing four prevalent virulence genes (*iucA*, *iroN*, *rmpA*, and *rmpA2*)⁵" – please note that the *iucA* and *iroN* genes are each part of complete operons that are required for aerobactin and salmochelin expression, respectively i.e. it is not correct to imply that just four virulence genes are important here.

3. Lines 66-68: "Our research indicates that the pK2044-like plasmid confers evolutionary benefits to ST11-KL64 by altering its genetic element, which offers insight into the widespread prevalence of ST11-KL64 CRKP in severe infections." I'm not quite sure what is meant by 'altering its genetic element' – similar to phrase in abstract. Suggest rephrasing.

4. Lines 94-95: It is stated that serotypes were identified with Kleborate – if Kaptive was switched on to generate full locus information (I assume it was if KL64/KL47 designations given) please also cite the relevant Kaptive paper.

5. Lines 108-109: "after removing a length of longest 109 contig > 4.5Mb and < 40 total contigs in the assembly." Should this be < 4.5 Mbp?

6. Line 224: "ST11 CRKP and Hv-CRKP grouped by serotype, especially in the KL64 clade." I don't understand this statement – how can resistance types group by serotype within a clade that is defined by the KL64 capsule locus (proxy for serotype)?

7. Lines 227-229: “Those differences implied that both virulence plasmids and antimicrobial resistant plasmids had a tendency to select the ST11-CRKP host to enter and stably exist. I don’t understand the logic here, please expand.

8. Lines 257-259: “For example, in ST11-KL64 strains, the gene g142 (957 bp) encoding the DsbA family protein, which was essential for bacterial virulence factor assembly ($p < 0.001$).” Does this p-value relate to a tested association between KL64 and this gene? Suggest rephrasing here as the current sentence can be read as if the p-value relates to the confidence that the gene is associated with virulence factor assembly – which has not been tested in this study as far as I can see.

9. Lines 371-389: Discussion of the impact of the plasmid deletion region on antioxidant capacity and macrophage survival centres on the impact of the reversal mutant i.e. reintroduction of the deleted region in the ST11 KL64 background. Did the authors also test the wt ST23 and knockout mutants? Would we expect to see the opposite trend?

10. Line 397-400: “These insights regarding the evolution of virulence plasmids could offer novel perspectives on the virulence mechanism of ST11 Hv-CRKP and partially account for the inability of phenotypes and genotypes to corroborate each other.” I’m not quite sure what is meant by the last part of this sentence. Suggest expanding or rephrasing the text.

11. Lines 409-411: “However, that was not a rigorous conclusion in genetics because they used a mapping approach to an assembly (generated using hybrid sequencing) which therefore does include the accessory genome to some extent in the phylogeny.” I disagree here – it may be appropriate to use a reference including ‘accessory’ gene content when the analysis is focusses specifically on a very closely related set of strains such as those representing a local clonal expansion, as these strains will likely share many accessory genes including those located on plasmids. In fact, inclusion of a greater proportion of the genome can result in a higher resolution and more accurate analysis in these cases. Please consider revising this critique or providing additional evidence from the data to support it.

12. Lines 437-442: “One hypothesis to explain the emergence of Hv-CRKP is that ST11 CRKP acquired virulence plasmids from sensitive but hypervirulent *K. pneumoniae* (hvKp) during transmission in hospital settings^{5,9}, but the virulence plasmid was often non-conjugative. Previous studies^{51,52} proposed that the existence of homologous regions between the two types of plasmids, it is putatively capable of co-integrated transfer with the help of conjugative KPC-2 plasmids.” I agree and note that there is in fact a published study demonstrating cotransfer in vitro that the authors may wish to cite here: Wang et al 2022 <https://journals.asm.org/doi/10.1128/spectrum.01364-22>.

Response to reviewers

Reviewer #1 (Remarks to the Author):

This is a solid study of the evolutionary of a *K. pneumoniae* subclone of ST11-KL64. The authors collected an impressive genome data set from online data bases and also data generated here. They convincingly demonstrate that this subclone acquired a virulence plasmid from from KL1 and that it is characterised by a major deletion.

I don't have major concerns with this work, but rather a few important suggestions to try to harness the power of such genomic data set.

- Please do not report p-values for root-to-tip regressions. The typical program used for this purpose, TempEst, does not report them because they are statistically invalid, due to pseudoreplication. Instead, just reporting the R^2 value would suffice. I also strongly suggest showing the regression. There are some *K. pneumoniae* data sets that have sufficient temporal signal and sometimes this is obvious from the regression.

Thank you for your valuable guidance. We appreciate your suggestion regarding the reporting of p-values for root-to-tip regressions. Considering this, we have removed p-values in line 220 and placed the regression (Supplementary Figure 3). Preliminary observations were conducted on the sample distribution of 239 samples in the entire dataset after excluding 7 samples without recorded collection date using TempEst. We observed that the early sequences belonging to serotypes KL125, KL105, KL103, etc., exhibited distant phylogenetic relationships compared to the rest of the dataset (a). Removal of these sequences resulted in an improved regression, with an R^2 value of 0.2552. It is important to note that this may still not represent a statistically robust temporal signal, likely due to the relative scarcity of early strains. Conducting grouping analysis based on serotypes may potentially yield improved correlations. However, this approach would not provide insights into the relationships among different serotype strains.

- The root-to-tip regression can sometimes be misleading. I suggest conducting a formal approach to assess temporal signal, BETS (Duchene et al. 2020; see also https://beast.community/bets_tutorial) that has much higher sensitivity. This test can be effectively conducted under a simple coalescent model and I recommend setting a small prior on the population size (for example a gamma prior distribution (shape = 0.001; scale = 1,000.0)). If this analysis detects temporal signal, then it would be very interesting to tease out the evolutionary timescale of these lineages. I do understand that this entails additional analyses, so I leave it up to the authors whether they want to include this point.

Like you, we are also eager to explore the evolutionary timescale of these lineages. We used BETS to assess the temporal signal and made several attempts to evaluate the time signal by excluding potentially discrepant samples from the dataset, despite of the high

computational cost. After removing outliers, a dataset comprising of 218 sequences was generated. In root-to-tip regressions, a notable improvement in model fit is observed, indicated by an increased R-squared value of 0.25. Using BETS, initial evidence supporting the presence of temporal signal has begun to emerge, with a log Bayes factor of 104 (Supplementary Table 3). However, the strict clock hypothesis is preferred over the UCLN (uncorrelated relaxed clock) hypothesis, possibly due to a scarcity of available early data and the disproportionate concentration of existing data in certain years. Taking this into consideration, we have included the standard phylogenetic analysis in the main manuscript, while the dated phylogenetic tree has been presented in the Supplementary. Perhaps in the future, with access to larger datasets and the opportunity for scientifically guided sampling strategies, a more accurate estimation of the evolutionary timescale can be obtained.

- Why were samples with no collection times excluded, particularly given that they were not used to calibrate the molecular clock? Standard phylogenetic analyses (i.e. not using a molecular clock), are agnostic about the timescale. Moreover, if it is only a couple of samples, their age can be estimated under a Bayesian approach (https://beast.community/tip_date_sampling).

All 246 samples, including those without collection dates, are included for standard phylogenetic analyses now. The new tree has been reconstructed and is depicted in Figure 1 of the revised edition, replacing the old tree. For samples with completely unknown collection date, the Bayesian approach appears to be unable to make inferences, as they lack even the year of collection. However, considering the limited impact of these no date samples, which pertains to only seven samples in total, its influence on the overall findings is likely negligible.

- Finally, it is very common practice to exclude recombining regions from bacterial genome data sets. However, it is important to report what proportion of the alignment was removed due to recombination. For example, what was the longest stretch removed from a sequence, or what was the average stretch removed across the alignment. I find this very useful to understand how much information could have been lost in recombination removal.

I have added the lengths of post-recombination alignments and recombinant regions in the method of the manuscript (lines 110-112). Furthermore, in the Supplementary Figure 2, I have included the positional information of the recombination regions in the alignment. Evidently, a significant proportion of recombination regions is observed across the global samples, encompassing diverse serotypes.

Duchene, S., Lemey, P., Stadler, T., Ho, S. Y., Duchene, D. A., Dhanasekaran, V., & Baele, G. (2020). Bayesian evaluation of temporal signal in measurably evolving populations. *Molecular Biology and Evolution*, 37(11), 3363-3379.

Reviewer #2 (Remarks to the Author):

The manuscript by Ruobing Wang and colleagues describes a functional genomics analysis of carbapenem-resistant and potentially hypervirulent *Klebsiella pneumoniae* (Kp). It is now well documented that the dominant carbapenem-resistant Kp lineage disseminated in China is sequence type 11 carrying the KPC-2 carbapenemase. Initially, dissemination of this clone was associated with the KL47 capsule synthesis locus but it is now clear that a recombinant derivative carrying the KL64 locus has displaced the KL47 progenitor. Worryingly, from a public health perspective, these multi-drug-resistant ST11-KL64 strains have also acquired a virulence plasmid carrying the key genes associated with hypervirulence, and which has been shown to increase virulence in mouse/*Galleria* models and increase patient mortality. Hence any works seeking to better understand the evolution and factors contributing to the widespread dissemination of these Kp are of high importance to the *Klebsiella* research and public health research communities.

Here the authors use genomic comparisons to compare the virulence plasmid acquired by ST11-KL64 to those carried by other ST11 and unrelated Kp e.g. ST23. They show i) that the ST11-KL64 plasmid is closely related to that carried by ST23, and ii) that ST11-KL64 strains are associated with a conserved deletion of ~3Kb in the virulence plasmid. Subsequently, RNASeq is used to explore the impact of the 3Kbp region on gene expression. The authors identify methionine metabolism and oxidative stress pathway genes enriched among those that are differentially expressed between ST11 wild type strains and their isogenic mutants with plasmid deletion reversal, as well as among ST23 wild type strains (with intact virulence plasmid) vs their isogenic plasmid deletion mutants. Finally, the authors demonstrate that the ST11 plasmid deletion reversal mutant has reduced antioxidant capacity and reduced macrophage survival compared to the isogenic wild type strain. The study presents a novel biological hypothesis for the rapid expansion of carbapenem-resistant and virulent ST11 Kp carrying the KL64 capsule locus- which will likely be of broad interest to the *Klebsiella* research community. While the general approach as described is sound, there are many missing methodological details – particularly regarding the RNASeq analysis (see below). Additionally, the results section is drastically lacking in quantitative description (see below examples) and the underlying data are not available (neither sequence data nor processed RNASeq analysis data) so it is not possible to fully assess the quality of this evidence. Additionally, the conclusion that the 3Kb deletion has contributed to the success of ST11-KL64 (e.g. as opposed to ST11-KL47) is somewhat circumstantial at this stage because there is no direct comparison of RNASeq nor oxidative stress / macrophage survival between ST11-KL64 with /without the deletion and ST11-KL47.

Finally, regarding the findings about the evolutionary origins of the ST11-KL64 virulence plasmid e.g. as stated at lines 418-423: “We discovered a potential link between the plasmids and the chromosome and hypothesized that pK2044-like virulence plasmids transfer into ST11- KL64 CRKP from ST23-KL1, even though it was challenging to

determine when the virulence plasmid movement happened. In a word, virulence plasmids from ST23-KL1 were acquired and independently evolved in ST11-KL64, and then spread for the past decades.” I agree this is very likely the case and is supported by the data presented here. However, this is also something that I would expect given the data that have already been reported in the literature about the similarity between ST11-KL64 virulence plasmids and pK2044 (e.g. Ruan et al *Infect Drug Resist.* 2020; 13: 199–206. Guo et al *Front Microbiol.* 2022; 13: 929826. etc), as well as what is known about the diversity of virulence genes / plasmids generally (see Lam et al *Genome Med* 2018 Oct 29;10(1):77. doi: 10.1186/s13073-018-0587-5, Spader et al. *Genome Med* 2023 Jan 19;15(1):3.doi: 10.1186/s13073-023-01153-y.) and the dominance of ST23 among hvKp, including in China (e.g. see Liu et al *Virulence* 2020 Dec;11(1):1215-1224.). It would be pertinent to include a discussion of the existing literature to contextualise the findings described here and tone down the originality claims.

Thank you for your meticulous review, which greatly contributed to the professionalism and rigor of my manuscript. Initially, due to a limited number of words, I provided only a brief overview of the methodology. Now, I will include the details in the method. Furthermore, we did additional *in vitro* or *in vivo* antioxidant experiments to compare the antioxidant capacity of KL64 (with the deletion) and KL47 (without the deletion) hv-CRKP strains (Supplementary Figure 6b). Additionally, we have tempered the originality claim regarding the phylogenetic analysis section.

MISSING METHODOLOGICAL DETAILS

1. Please describe the isolate collection from which the isolates in the current study were derived. There is a description of an ethics approval but no description of the study or collection.

The strains collected in this study were derived from several long-term Chinese multicenter surveillances conducted by our team, as well as strains collected in previous studies. Most of isolates were from the CRE-network and Chinese Antimicrobial Resistance Surveillance of Nosocomial Infections (CARES) project, which collected strains from hospitals in various provinces and cities in China for antimicrobial susceptibility testing and sequencing of resistant or virulent strains. The sampled methods have been described in the previous articles. Based on these efforts, our team has established a sequencing database. The brief description of where these strains from is mentioned in lines 75-77, and their regional distribution can be found in Supplementary Figure 1. Additionally, the appendix contains information on the collection date, location, and specimen types of the strains used in the analysis.

2. DNA extraction and sequence library preparation methods should be provided for the new sequence data generate here. For the Illumina data please also indicate if single or paired end and state the read length.

The description of DNA extraction, library preparation and sequencing has been added in

line 80-83 of the revision.

3. RNA extraction and library prep and sequencing details (e.g. read lengths) should be stated. The approach for processing these data should also be stated – presumably some sort of referenced based approach was used? If so, which mapping algorithm, parameters, what reference sequences were used?

The details of transcriptome sequencing have been added in line 145-151 of the revision. The sequencing experiment was conducted by Shanghai Majorbio Bio-pharm Technology Co., Ltd, while the analysis was performed using the online platform of Majorbio Cloud Platform (www.majorbio.com, Ren, Y. et al. Majorbio Cloud: A one-stop, comprehensive bioinformatic platform for multiomics analyses. *iMeta* 1, e12 (2022).).

4. Please indicate if any read filtering or QC steps were undertaken prior to genome assembly and on the assemblies themselves (including those downloaded from NCBI).

For the new sequencing data, quality control was performed using the default parameters of fastp to obtain clean data, which was then subjected to hybrid assembly. It has been described in line 84-85 of the revision. As for the assemblies from public database, we utilized them from the NCBI Reference Sequence database (claim to be a comprehensive, integrated, non-redundant, well-annotated set of reference sequences) and selected those with a longest contig > 4.5 Mb for chromosomal analysis.

5. Genome accessions should be provided. Both the project IDs and the individual isolate / experiment accessions must be provided for the purposes of reproducibility.

The genomic data has been uploaded in the NCBI database and will be released upon publication of the study. The BioProject accession number is provided in line 87, and the numbers for each strain will be included in the supplementary table.

6. Lines 85-90: “that were sufficiently contiguous to allow resolution of chromosome and plasmid, to investigate global patterns of CRKP and contextualize our data. We set out to identify, from this dataset, sufficiently contiguous assemblies for resolution of bacterial chromosome and plasmid. Following filtering according to those criteria, a total of 1,219 assemblies were considered from the NCBI Reference Sequence database (n = 917) and our database (n = 302).” The specific criteria should be stated.

Because short read sequences on the Illumina platform typically do not allow easy assignment of contigs to plasmid or chromosomal DNA. The chromosomal genome of *K. pneumoniae* is usually around 5 Mb in size, with several plasmids. Therefore, criteria were devised to select publicly available assemblies long enough to be studied, including 1) longest contig length > 4.5 Mb, and 2) number of contigs < 40. The criteria have been added in line 93-96 of the revision.

7. Lines 97-99: “Please indicate any parameters or thresholds used for structural analyses

and to filter UniProt or FoldSeek outputs.

We performed structural analyses and filtered FoldSeek using default parameters and thresholds.

8. Line 101: Please indicate which type of BLAST (e.g. BLASTn, tBLASTn) and thresholds used to confirm virulence gene locations.

We extracted sequence of virulence genes from pLVPK (AY378100.1) and pK2044 (NC_006625.1), and BLASTn with using default parameters. Most of the results showed a coverage of greater than 90% and an identity of greater than 97% without any filtering. Except for *iroN*, over 20% of the sequences only aligned with the first 50% of the gene. Therefore, the incomplete alignment of *iroN* was marked with *iroN**. This description has been added to line 109-110.

9. Lines 102-104: "We selected 103 virulence contigs from 201 isolates in the predominant serotypes KL1, KL2, KL47, or KL64 to analyze." What criteria were used to select these sequences?

Sorry for any confusion. What I meant is that out of the 217 virulence contigs, they were derived from 201 strains. Among the 1,219 assemblies, a total of 258 sequences were identified as hvKP based on virulence score > 2 using Kleborate. However, the tool did not determine the virulence contigs, which hindered subsequent evolutionary analysis. Hence, five common virulence-associated genes were used to locate the virulence contigs (plasmids) by BLASTn. Given that KL1, KL2, KL47, and KL64 correspond to the most prevalent serotypes in hypervirulent or ST11 *K. pneumoniae*, particular attention was given to the evolutionary analysis of virulence contigs in these four serotypes. Finally, 217 virulence sequences were obtained from the 201 strains belonging to these four serotypes. The description of this section was revised in line 111-115 of the revision to provide a more precise and clear explanation.

10. Line 110-111: Please specify the parameters that were used to run Roary and the definition of core i.e. >= 95%, 100% etc. Also state any parameters for ClonalFrameML. What was the length of the sequence alignment used for ClonalFrame analysis and how much sequence was masked by this analysis?

We employed Roary (<https://github.com/sanger-pathogens/Roary#usage>) to obtain the multiple sequence alignment (MSA) with basic parameters as recommended in its official documentation. The core gene (99% <= strains <= 100%) was directly derived from Roary, which generated the core gene alignment for initial phylogenetic tree construction. A total of 55,262 bp of recombination regions were identified using ClonalframeML with emsim 100 and basic parameters as recommended (<https://github.com/xavierdidelot/clonalframeml/wiki>) and removed from the MSA, resulting in a final alignment of 758,264 bp. The length of the identified recombination regions and the final MSA are now updated in lines 121 and 123 of the revision.

11. Lines 115-116: "we focused on plasmids with high sequence identity (a coverage of >

95%) to the classic virulence plasmid pK2044 (NC_006625.1)” Should this indicate that you focussed on plasmids with high sequence coverage as specified within the parenthesis? Or if both identity and coverage cut-offs were applied, please indicate both here.

The updated description regarding the calculation of ANI and the definition of highly similar contigs, where high similarity is defined as ANI > 95 and coverage > 80%, can be found in line 126-130 of the revision.

12. Lines 118-120: “Correlation between genes annotated on identified virulence plasmids and discrete phenotypic characteristics were analyzed using Scoary and verified using BLAST.” Which phenotypic characteristics were tested and how were these defined? Please also state any parameters for Scoary and indicate BLAST type/thresholds.

Serotypes, as a phenotypic characteristic, was employed to investigate whether different serotypes exhibit genetic variations in virulence contigs. The analysis was conducted using Scoary with the standard parameters provided by the official recommend (<https://github.com/AdmiralenOla/Scoary#usage>). Subsequently, the identified specific genes were subjected to secondary confirmation using BLASTn. The BLASTn analysis focused primarily on the identity, typically above 95%, while also considering the coverage. Genes with low coverage were represented in lighter shades in the Figure 3 of the revision. The description has also been revised in the new version.

13. Lines 120-121:” The final plasmid phylogenetic tree was visualized in R. – Please name and cite the appropriate R packages.

The reference to the R package has been added at line 135 of the revision.

14. Lines 125-126: “was identified as consistently absent when the plasmid was associated with ST11 CRKP across our datasets.” How was absence determined? (如何确定不存在-blast)

As stated in the response to question 12, we utilized BLASTn to determine the complete or partial presence of genes by considering both identity and coverage. In the output of the BLASTn, identities were usually at above 90%, while coverage played a crucial role in determining the completeness of a gene. Here, the term “absence” refers to the lack of any results obtained when employing BLASTn with the standard parameters.

15. Lines 137-139:” RNA sequencing was performed with three replicates to quantify patterns of gene 138 expression in isolates C4599, C4599Δ3K, C1789::pEASY, C1789::hp1-4, and 139 C1789::hp5 to better understand the role of the ~3 Kb region.” Why were these strains chosen and not others? E.g. Why wasn’t TJ12019 and its mutant derivatives as described in lines 126-130 used?

We selected these two strains based on their possession of pK2044-like virulence plasmids, which have the highest average nucleotide identity (ANI) to pK2044 and the classic KPC-

2 plasmid. C1789 represents the ST11-KL64 Hv-CRKP strain, while C4599 represents the ST23-KL1 Hv-CRKP strain. TJ12019, on the other hand, is a ST23-KL1 HvKP strain that does not carry the KPC-2 plasmid. Our goal is to maintain the virulence plasmid in the same environment, including the host and additional plasmids, in order to minimize any potential impacts caused by other plasmids. We added the explanation in line 142 and 153-155 of the revision.

16. Line 143: "The Majorbio Cloud Platform (www.majorbio.com) and R were used to analyze the data." Please state and cite the R package(s) used.

The reference to the R package has been added at line 135 of the revision.

17. Lines 201-202: "however, none of these isolates harbored virulence plasmids based on the presence of virulence associated marker genes." Which marker genes were used?

We used five common virulence-associated genes, *iucA*, *iroN*, *rmpA*, *rmpA2* and *pagO*, as markers. They were mentioned in line 107-108 in method section.

18. Lines 218-219: "However, because of the weak temporal signal in the data ($R^2 = 0.12$, $P = 0.0001$)," How was temporal signal determined?

The temporal signal was evaluated using Tempest and the Bayesian approach (http://beast.community/tempest_tutorial, http://beast.community/bets_tutorial). The temporal signal result was showed in Supplementary Figure 3.

19. Line 222: "How was 'closely related' defined?"

Sorry for the confusion. What I meant is that they are located on the adjacent branch of the phylogenetic tree. I changed the sentence in the line 242 of the revision.

LACK OF QUANTITATIVE DESCRIPTION

1. Line 225-227: "The virulence plasmid in KL64 strains were more similar to the classic virulence plasmid pK2044, while KL47 strains carried fewer virulence genes integrated into the chromosome." How was similarity to the pK2044 plasmid defined here? In terms of total number of virulence genes carried on the plasmid?

Sorry for the confusion. What I meant is that the virulence plasmid in KL64 strains were derived from a branch of KL1 and closer to the classic virulence plasmid pK2044 in terms of phylogenetic relationship. As for the 'similarity', we defined it by calculate the ANI between the pK2044 and virulence plasmids and the coverage of virulence plasmids mapped to pK2044. The paragraph has been rephrased.

2. Line 307-308: Discussion of differentially expressed genes. Please describe a summary

of the results e.g. total number of genes up / downregulated, and fold change ranges etc, and give specific fold change / p values for the genes discussed. These details are provided in the supplementary figure but given the relevance to the key findings it is important to state clearly here as well.

The summary of differentially expressed genes is presented in lines 349-353. The fold change and p-values are also added in parentheses following the mentioned genes.

3. Lines 321-327: Do these lines refer to chromosomally encoded genes? Where are these data shown? Figure S4 appears to show only plasmid-encoded genes. Again, please give the specific details about the fold changes and p-values associated with these genes so that readers and reviewers can interpret and assess the findings.

This paragraph describes a summary of the transcriptome in C4599 group, with specific data available in Supplementary Figure 4c, which represents the results of total mRNA analysis. The fold change and p-values are also added in parentheses following the mentioned genes.

4. Lines 329-330: "Enrichment of both analyses indicated cysteine and methionine metabolism to be the most affected with 12 upregulated genes in C4599 Δ 3K (Figure 6A)." Please explain and provide quantifiable evidence to support this statement, In Fig 6A (KEGG enrichment analysis) I see that the point labelled for cysteine and methionine metabolism has the highest log fold change and highest enrich factor so this make sense. However, in 6B (GO analysis) I don't see any points labelled for cysteine metabolism and the methionine point has neither the highest fold change nor the highest enrich factor. Is there some reason that the other points should be discarded? Also please provide references for the information about the methionine pathway that is provided and shown in Figure S5.

The previous statement was not entirely accurate. Only the KEGG analysis revealed that the deletion is implicated in cysteine and methionine metabolism. On the other hand, GO analysis provides insights into the gene functions, pointing towards oxidoreductase activity. These two analyses have different focuses. The specific DEG data is presented in Supplementary Figure 5, along with the reference sources for the metabolic pathways.

5. Lines 341-353: - Discussion of up/down regulated genes in different mutants is missing quantitative evidence in the text and the figure shows no indication of a consideration of p-values or other appropriate approaches to exclude experimental noise.

The content of this section has been revised and added with quantitative evidence.

OTHER MINOR COMMENTS

1. Line 31-33: "The pK2044-like virulence plasmid provides ST11-KL64 with evolutionary

advantages through altering its genetic material to enhance survival.” I don’t see how this statement is supported by the data presented in this work. Suggest expanding or rephrasing.

The sentence has been rephrased.

2. Lines 42-45: “Over the past decade, hv-CRKP has rapidly disseminated, a phenomenon typically attributed to the acquisition 44 of virulence plasmids containing four prevalent virulence genes (*iucA*, *iroN*, *rmpA*, and *rmpA2*)⁵” – please note that the *iucA* and *iroN* genes are each part of complete operons that are required for aerobactin and salmochelin expression, respectively i.e. it is not correct to imply that just four virulence genes are important here.

The sentence has been rephrased.

3. Lines 66-68: “Our research indicates that the pK2044-like plasmid confers evolutionary benefits to ST11-KL64 by altering its genetic element, which offers insight into the widespread prevalence of ST11-KL64 CRKP in severe infections.” I’m not quite sure what is meant by ‘altering its genetic element’ – similar to phrase in abstract. Suggest rephrasing.

The sentence has been rephrased.

4. Lines 94-95: It is stated that serotypes were identified with Kleborate – if Kaptive was switched on to generate full locus information (I assume it was if KL64/KL47 designations given) please also cite the relevant Kaptive paper.

We have incorporated the citation of Kaptive.

5. Lines 108-109: “after removing a length of longest 109 contig > 4.5Mb and < 40 total contigs in the assembly.” Should this be < 4.5 Mbp?

This error has been corrected.

6. Line 224: “ST11 CRKP and Hv-CRKP grouped by serotype, especially in the KL64 clade.” I don’t understand this statement – how can resistance types group by serotype within a clade that is defined by the KL64 capsule locus (proxy for serotype)?

Response: Sorry for not expressing clearly. The sentence has been rephrased.

7. Lines 227-229: “Those differences implied that both virulence plasmids and antimicrobial resistant plasmids had a tendency to select the ST11-CRKP host to enter and stably exist. I don’t understand the logic here, please expand.

The sentence has been rephrased.

8. Lines 257-259: “For example, in ST11-KL64 strains, the gene g142 (957 bp) encoding the DsbA family protein, which was essential for bacterial virulence factor assembly ($p < 0.001$).” Does this p-value relate to a tested association between KL64 and this gene? Suggest rephrasing here as the current sentence can be read as if the p-value relates to the confidence that the gene is associated with virulence factor assembly – which has not been tested in this study as far as I can see.

Sorry for the confusion. The p-value mentioned refers to the correlation test between the gene g142 and KL64, not the association with virulence factor assembly. The revised version has been amended to remove the p-value accordingly.

9. Lines 371-389: Discussion of the impact of the plasmid deletion region on antioxidant capacity and macrophage survival centres on the impact of the reversal mutant i.e. reintroduction of the deleted region in the ST11 KL64 background. Did the authors also test the wt ST23 and knockout mutants? Would we expect to see the opposite trend?

We conducted experiments to assess intracellular survival of the ST23 wildtype strain and its mutants in macrophages, as described in Supplementary Figure 6a. The results revealed a trend contrary to that observed in the ST11 KL64 background. The knockout of the deletion region led to an increased, albeit non-significant, survival rate, whereas the reversed mutants exhibited a significant decrease in survival rates.

10. Line 397-400: “These insights regarding the evolution of virulence plasmids could offer novel perspectives on the virulence mechanism of ST11 Hv-CRKP and partially account for the inability of phenotypes and genotypes to corroborate each other.” I’m not quite sure what is meant by the last part of this sentence. Suggest expanding or rephrasing the text.

The sentence has been rephrased.

11. Lines 409-411: “However, that was not a rigorous conclusion in genetics because they used a mapping approach to an assembly (generated using hybrid sequencing) which therefore does include the accessory genome to some extent in the phylogeny.” I disagree here – it may be appropriate to use a reference including ‘accessory’ gene content when the analysis is focusses specifically on a very closely related set of strains such as those representing a local clonal expansion, as these strains will likely share many accessory genes including those located on plasmids. In fact, inclusion of a greater proportion of the genome can result in a higher resolution and more accurate analysis in these cases. Please consider revising this critique or providing additional evidence from the data to support it.

You make a valid point, and I have made amendments to this statement.

12. Lines 437-442: “One hypothesis to explain the emergence of Hv-CRKP is that ST11 CRKP acquired virulence plasmids from sensitive but hypervirulent *K. pneumoniae* (hvKp)

during transmission in hospital settings^{5,9}, but the virulence plasmid was often non-conjugative. Previous studies^{51,52} proposed that the existence of homologous regions between the two types of plasmids, it is putatively capable of co-integrated transfer with the help of conjugative KPC-2 plasmids.” I agree and note that there is in fact a published study demonstrating cotransfer in vitro that the authors may wish to cite here: Wang et al 2022 <https://journals.asm.org/doi/10.1128/spectrum.01364-22>.

Thank you for providing the reference. The paragraph has been rephrased.

REVIEWER COMMENTS

Reviewer #1 (Remarks to the Author):

The authors have addressed all my comments and concerns. I have no further issues here. I would be happy to recommend this manuscript for publication, but I do need to highlight that the other reviewer made some important comments that are beyond my expertise and which should be addressed prior to its actual acceptance.

Reviewer #2 (Remarks to the Author):

The revised manuscript is improved and the authors have addressed the major concerns, and most but not all of the additional concerns. In several instances, the concerns about missing methodological details are addressed in the response but no changes appear to have been made to the text – which is not helpful for future readers of the paper. However,

I note that it was extremely difficult to fully assess this revision because;

- a) the authors have not noted the specific manuscript changes in the responses document;
- b) the line numbers given in the response document do not match where the relevant content is in the revised PDF;
- c) in several places the written response does not match the changes in the text.

I have noted the outstanding points below.

REV 2 GENERAL COMMENT

“We discovered a potential link between the plasmids and the chromosome and hypothesized that pK2044-like virulence plasmids transfer into ST11- KL64 CRKP from ST23-KL1, even though it was challenging to determine when the virulence plasmid movement happened. In a word, virulence plasmids from ST23-KL1 were acquired and independently evolved in ST11-KL64, and then spread for the past decades.” I agree this is very likely the case and is supported by the data presented here. However, this is also something that I would expect given the data that have already been reported in the literature about the similarity between ST11-KL64 virulence plasmids and pK2044 (e.g. Ruan et al Infect Drug Resist. 2020; 13: 199–206. Guo et al Front Microbiol. 2022; 13: 929826. etc), as well as what is

known about the diversity of virulence genes / plasmids generally (see Lam et al Genome Med 2018 Oct 29;10(1):77. doi: 10.1186/s13073-018-0587-5, Spader et al. Genome Med 2023 Jan 19;15(1):3.doi: 10.1186/s13073-023-01153-y.) and the dominance of ST23 among hvKp, including in China (e.g. see Liu et al Virulence 2020 Dec;11(1):1215-1224.). It would be pertinent to include a discussion of the existing literature to contextualise the findings described here and tone down the originality claims.”

The novelty claims have been toned down but there is no discussion about the existing literature. Please respect the body of work that is already available and aid your readers by contextualising your findings.

REV 2 MISSING METHODS DETAILS

Point 1. Please describe the isolate collection from which the isolates in the current study were derived. There is a description of an ethics approval but no description of the study or collection.

Thanks for clarification, please briefly add this information to the manuscript to help your readers,

Point 3. RNA extraction and library prep and sequencing details (e.g. read lengths) should be stated. The approach for processing these data should also be stated – presumably some sort of referenced based approach was used? If so, which mapping algorithm, parameters, what reference sequences were used?

I still don't see a description of the mapping algorithm, parameters and reference sequence that was used.

Point 4. Please indicate if any read filtering or QC steps were undertaken prior to genome assembly and on the assemblies themselves (including those downloaded from NCBI).

The information provided in the response (use of fastp) does not match that in the updated text (trimmomatic), please confirm and update the text if necessary.

Point 8. Line 101: Please indicate which type of BLAST (e.g. BLASTn, tBLASTn) and thresholds used to confirm virulence gene locations.

The values stated in the response do not match those in the updated text. Please confirm and update the text accordingly.

Point 10. Line 110-111: Please specify the parameters that were used to run Roary and the definition of core i.e. $\geq 95\%$, 100% etc. Also state any parameters for ClonalFrameML. What was the length of the sequence alignment used for ClonalFrame analysis and how much sequence was masked by this analysis?

The response mentions that default parameters were used but this does not seem to have been added to the text. Please add to the text so that future readers can replicate your work.

Point 12. Lines 118-120: "Correlation between genes annotated on identified virulence plasmids and discrete phenotypic characteristics were analyzed using Scoary and verified using BLAST." Which phenotypic characteristics were tested and how were these defined? Please also state any parameters for Scoary and indicate BLAST type/thresholds.

The response describes the details but again these are not added to the text, specifically the parameters and BLASTn thresholds. Please update the text for the benefit of readers, not just reviewers.

Point 14. Lines 125-126: "was identified as consistently absent when the plasmid was associated with ST11 CRKP across our datasets." How was absence determined?

Again, the response is given but the details should be added to the manuscript.

Response to reviewers

We would like to express our gratitude for arranging the peer review of our manuscript and providing valuable feedback and suggestions. We greatly appreciate the guidance and support of both reviewers, as their expertise has been instrumental in inspiring our research work.

In response to each of the reviewer's comments, we have carefully considered and offer the following responses shown in blue immediately underneath the reviewers' comments. Where we reference line numbers, these refer to the resubmitted **clean version** of the manuscript. To make it easier to find the changed content in the manuscript, the lines mentioned in the response have been highlighted in the PDF document.

Thank you for your understanding and for giving us the opportunity to enhance the quality of our manuscript.

Sincerely yours,

Ruobing Wang and Hui Wang, on behalf of all the co-authors

Reviewer #1 (Remarks to the Author):

The authors have addressed all my comments and concerns. I have no further issues here. I would be happy to recommend this manuscript for publication, but I do need to highlight that the other reviewer made some important comments that are beyond my expertise and which should be addressed prior to its actual acceptance.

Thank you for your thorough review of our manuscript and recommendation for publication.

Reviewer #2 (Remarks to the Author):

The revised manuscript is improved and the authors have addressed the major concerns, and most but not all of the additional concerns. In several instances, the concerns about missing methodological details are addressed in the response but no changes appear to have been made to the text – which is not helpful for future readers of the paper. However, I note that it was extremely difficult to fully assess this revision because;

- the authors have not noted the specific manuscript changes in the responses document;
- the line numbers given in the response document do not match where the relevant content is in the revised PDF;
- in several places the written response does not match the changes in the text.

I have noted the outstanding points below.

Thank you for taking the time to review our revised manuscript. We appreciate your

feedback and have carefully considered your comments. Sorry for the inconvenience caused by the issues you encountered during your assessment of the revision. It is possible that on one hand, we have moved the method section to the end before submission, and on the other hand, the location and context of the changes were not displayed in the revised PDF version.

We will rectify this error in the revised version by ensuring accurate alignment between line numbers and content and carefully review and align our responses with the actual modifications made to the manuscript. To facilitate the review process of the modifications, we also highlighted sentences within the PDF document.

REV 2 GENERAL COMMENT

“We discovered a potential link between the plasmids and the chromosome and hypothesized that pK2044-like virulence plasmids transfer into ST11- KL64 CRKP from ST23-KL1, even though it was challenging to determine when the virulence plasmid movement happened. In a word, virulence plasmids from ST23-KL1 were acquired and independently evolved in ST11-KL64, and then spread for the past decades.” I agree this is very likely the case and is supported by the data presented here. However, this is also something that I would expect given the data that have already been reported in the literature about

the similarity between ST11-KL64 virulence plasmids and pK2044 (e.g. Ruan et al *Infect Drug Resist.* 2020; 13: 199–206. Guo et al *Front Microbiol.* 2022; 13: 929826. etc), as well as what is known about the diversity of virulence genes / plasmids generally (see Lam et al *Genome Med* 2018 Oct 29;10(1):77. doi: 10.1186/s13073-018-0587-5, Spader et al. *Genome Med* 2023 Jan 19;15(1):3.doi: 10.1186/s13073-023-01153-y.) and the dominance of ST23 among hvKp, including in China (e.g. see Liu et al *Virulence* 2020 Dec;11(1):1215-1224.). It would be pertinent to include a discussion of the existing literature to contextualise the findings described here and tone down the originality claims.”

The novelty claims have been toned down but there is no discussion about the existing literature. Please respect the body of work that is already available and aid your readers by contextualising your findings.

Sorry for not including these articles previously, as we were mindful of the reference limit. In light of your suggestion, we have restructured the manuscript and incorporate a discussion of this content into the end of second paragraph of Discussion in the revision (line 303-311).

REV 2 MISSING METHODS DETAILS

Point 1. Please describe the isolate collection from which the isolates in the current study were derived. There is a description of an ethics approval but no description of the study or collection.

Thanks for clarification, please briefly add this information to the manuscript to help your readers.

We have added this information to the revision (line 372-373) and in a designated section of the supplementary methods.

Point 3. RNA extraction and library prep and sequencing details (e.g. read lengths) should be stated. The approach for processing these data should also be stated – presumably some sort of referenced based approach was used? If so, which mapping algorithm, parameters, what reference sequences were used?

I still don't see a description of the mapping algorithm, parameters and reference sequence that was used.

C1789 and C4599 were used as references for their respective groups, and the cleaned reads were aligned to the reference using Bowtie 2 with default parameters. We have added this information to 'Transcriptome sequencing and analysis' of the supplementary methods.

Point 4. Please indicate if any read filtering or QC steps were undertaken prior to genome assembly and on the assemblies themselves (including those downloaded from NCBI).

The information provided in the response (use of fastp) does not match that in the updated text (trimmomatic), please confirm and update the text if necessary.

Sorry for the confusion caused. For the newly sequenced strains in the study, genome data cleaning was performed using Trimmomatic and fastp at different periods due to the study spanning several years. As for RNAseq, SeqPrep and Sickle are used to filter data. We have updated the description in the second paragraph of 'Isolates collection and Genome sequencing' and the first paragraph of 'Transcriptome sequencing and analysis' of the supplementary methods.

Point 8. Line 101: Please indicate which type of BLAST (e.g. BLASTn, tBLASTn) and thresholds used to confirm virulence gene locations.

The values stated in the response do not match those in the updated text. Please confirm and update the text accordingly.

Thank you for pointing out this issue. We have updated the description in the revision accordingly (line 401-402).

Point 10. Line 110-111: Please specify the parameters that were used to run Roary and the definition of core i.e. $\geq 95\%$, 100% etc. Also state any parameters for ClonalFrameML. What was the length of the sequence alignment used for ClonalFrame analysis and how

much sequence was masked by this analysis?

The response mentions that default parameters were used but this does not seem to have been added to the text. Please add to the text so that future readers can replicate your work.

The parameters have been incorporated into the revised manuscript (line 411-416), and the description of this section has been rewritten accordingly.

Point 12. Lines 118-120: "Correlation between genes annotated on identified virulence plasmids and discrete phenotypic characteristics were analyzed using Scoary and verified using BLAST." Which phenotypic characteristics were tested and how were these defined? Please also state any parameters for Scoary and indicate BLAST type/thresholds.

The response describes the details but again these are not added to the text, specifically the parameters and BLASTn thresholds. Please update the text for the benefit of readers, not just reviewers.

The information has been added to the revision and the description of this section has been rewritten accordingly (line 430-433).

Point 14. Lines 125-126: "was identified as consistently absent when the plasmid was associated with ST11 CRKP across our datasets." How was absence determined?

Again, the response is given but the details should be added to the manuscript.

We have added the information in the revision (line 437-440).

REVIEWERS' COMMENTS

Reviewer #2 (Remarks to the Author):

The updated manuscript addresses all of the outstanding concerns. I would like to thank the authors for making it very easy to understand and locate the changes within the manuscript files.